# Causally-Aware Information Bottleneck for Domain Adaptation

**Mohammad Ali Javidian**                                                      *javidianma@appstate.edu*
*Department of Computer Science*
*Appalachian State University*

**Reviewed on OpenReview:** *https://openreview.net/forum?id=TbcqPEgJ9z*

## Abstract

We study a common domain adaptation setting in causal systems with *local causal knowledge*: the target variable is observed in the source domain but is entirely missing in the target domain, and the conditional mechanism of the target given its Markov blanket is assumed stable across domains. We aim to impute the target variable in the target domain from the remaining observed variables under various shifts. Our central transfer mechanism is structural: restricting the predictor to the Markov blanket of the target screens off shift-prone non-blanket variation and yields zero-shot transfer under blanket invariance. On top of this restriction we frame estimation as learning a compact, mechanism-stable representation, and we instantiate it with the Information Bottleneck (IB) as a principled compression and regularization mechanism. For linear Gaussian causal models, we derive a closed-form Gaussian Information Bottleneck (GIB) solution that reduces to a canonical correlation analysis (CCA)–style projection and is provably lossless relative to using all non-target variables; in this well-specified regime, ordinary least squares on the blanket is already near-optimal, so the value of IB is regularization rather than accuracy gains. For nonlinear or non-Gaussian data, where no closed-form conditional estimator is available, we introduce a Variational Information Bottleneck (VIB) encoder–predictor that scales to high dimensions and can be trained on source data and deployed zero-shot to the target domain. Across synthetic and real datasets, our approach consistently attains accurate imputations, supporting practical use in high-dimensional causal models and furnishing a unified, lightweight toolkit for causal domain adaptation.

## 1 Introduction

Modern prediction systems are rarely deployed in the same environment in which they were trained. Shifts in demographics, sensors, or policies alter the joint distribution of variables and erode accuracy. This motivates *domain adaptation*: transferring knowledge from a labeled *source* to an unlabeled or partially labeled *target* whose distribution differs. In fact, when a model trained in one environment fails in another, the culprit is rarely randomness; it is a change in the data-generating *mechanisms*. Two archetypal shifts capture common failures: Under *covariate shift*, the context distribution changes while the conditional mechanism remains fixed, i.e., $P(X)$ (or $P_{\text{source}}(X) \neq P_{\text{target}}(X)$) varies but $P(Y \mid X)$ is invariant (Shimodaira, 2000; Sugiyama et al., 2008; Johansson et al., 2019). Under *target (label) shift*, the marginal $P(Y)$ differs whereas $P(X \mid Y)$ is stable (Storkey, 2009; Zhang et al., 2013; Lipton et al., 2018). Broader taxonomies and theoretical results can be found in (Redko et al., 2019). We focus on settings where the target variable is *missing in the target domain* and the goal is to impute it reliably under covariate, target, or constrained generalized target shift where the marginal $P(Y)$ changes, and the conditional $P(X|Y)$ changes with constraints.

**Causality as a stabilizer.** Causal structure provides a lens to separate robust signal from spurious correlation. In theory, causal inference tools aim to protect against instability by aligning with the graph of

cause and effect. For example, *selection diagrams* formalize differences between populations and support *transportability* via do-calculus (Pearl, 2009; Bareinboim & Pearl, 2011; 2012; 2014; Correa & Bareinboim, 2019). *Invariant causal prediction* (ICP) searches for subsets whose residuals are stable across environments (Peters et al., 2016; Pfister et al., 2019a;b), while *graph surgery* proactively removes unstable mechanisms (Subbaswamy & Saria, 2018; 2019). A complementary line frames adaptation as *graph pruning* to select predictors that yield invariant conditionals (Magliacane et al., 2018; Rojas-Carulla et al., 2018; Kouw & Loog, 2019). Despite their guarantees, these methods often require causal effect estimation or counterfactual reasoning, can be conservative (trading variance for zero transfer bias), and may struggle to scale. More recent advances in *invariant risk minimization* (Arjovsky et al., 2019) and deep generative approaches for causal representation learning (Krueger et al., 2021; Lv et al., 2022) promise improved robustness to unseen shifts.

**Our view: compact, mechanism-stable representations.** In this paper, we formulate adaptation as learning a *mechanism-stable* summary: a low-dimensional representation that retains target-relevant information and suppresses spurious variation. We design a *DAG-aware information bottleneck* that aligns the encoder with the causal structure while explicitly compressing the observed variables $X$ into a bottleneck $U$, which preserves valuable information for predicting $T$ and discards nuisance variation. For this purpose, we will use the DAG structure information (e.g., parents/Markov blanket) to focus the encoder on stable mechanisms and avoid unstable paths. This approach yields a representation that is small, causally grounded, and stable across domains, with formal guarantees in the Gaussian case and distribution-free justifications for the nonlinear/non-Gaussian regime.

**Motivating example (high-dimensional spurious block; MB-invariance holds).** We consider a linear–Gaussian SEM designed to isolate the benefit of restricting the bottleneck to MB($T$) under *domain shift that does not perturb the target mechanism*. The causal structure is shown in Fig. 1. Let $C = (C_1, \ldots, C_k)$ denote a small set of stable causal drivers of $T$, $S = (S_1, \ldots, S_m)$ a high-dimensional block of *spurious proxies* whose relationship to $C$ changes across domains, and $N = (N_1, \ldots, N_r)$ a nuisance block affected by the domain but irrelevant to $T$. In the *source* domain, the mechanisms are

$$C_i \sim \mathcal{N}(0, 1), \qquad i = 1, \ldots, k,$$
$$T = w^\top C + \varepsilon_T,$$
$$S_j = a_0 \left(w^\top C\right) + \varepsilon_{S_j}, \qquad j = 1, \ldots, m,$$
$$N_\ell = b_0 + \varepsilon_{N_\ell}, \qquad \ell = 1, \ldots, r,$$

with mutually independent noises $\varepsilon_T \sim \mathcal{N}(0, \sigma_T^2)$, $\varepsilon_{S_j} \sim \mathcal{N}(0, \sigma_S^2)$, $\varepsilon_{N_\ell} \sim \mathcal{N}(0, \sigma_N^2)$, and a fixed $w \in \mathbb{R}^k$. The Markov blanket of the target is therefore MB($T$) = $\{C_1, \ldots, C_k\}$, since $T$ has parents $C$ and no observed children or spouses in this example.

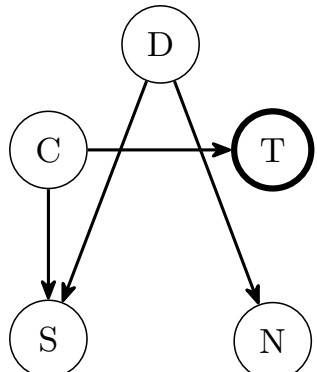

Figure 1: Causal DAG underlying the motivating example. **C**: stable causal drivers of $T$ (the Markov blanket); **S**: high-dimensional spurious proxy block whose conditional distribution shifts with **D**; **N**: nuisance block affected by **D** but irrelevant to $T$. The target mechanism $p(T \mid C)$ is invariant across domains.

**Shift scenario.** We consider a *large* shift in the target domain that changes only non-blanket mechanisms. Specifically, in the *target* domain we keep the target mechanism unchanged,

$$T = w^\top C + \varepsilon_T \qquad \text{(same } w, \sigma_T^2\text{)},$$

but we alter the proxy and nuisance blocks via domain-dependent parameters

$$S_j = a_1 \left( w^\top C \right) + \varepsilon_{S_j}, \qquad j = 1, \ldots, m,$$
$$N_\ell = b_1 + \varepsilon_{N_\ell}, \qquad \ell = 1, \ldots, r,$$

with $a_1 \neq a_0$ (in our experiments, a sign flip $a_0 = +1$, $a_1 = -1$) and $b_1 \neq b_0$. Crucially, this construction preserves

$$p_s(T \mid \mathrm{MB}(T)) \;=\; p_s(T \mid C) \;=\; p_t(T \mid C) \;=\; p_t(T \mid \mathrm{MB}(T)),$$

so the risk-transfer identity under MB-invariance applies, while the *global* input distribution shifts substantially.

As before, $T$ is unobserved at deployment in the target domain. We compare two approaches for predicting (imputing) $T$ under this shift: (*a*) the Markov blanket Gaussian Information Bottleneck (GIB), and (*b*) the global GIB. At a high level, a Gaussian Information Bottleneck learns a low-dimensional linear summary of its inputs that retains the directions most predictive of $T$ while compressing the rest; in the linear–Gaussian case this summary coincides with a canonical-correlation projection (formal definitions and the closed form are deferred to Sect. 4, specifically Sect. 4.1). The two variants differ only in which inputs the encoder sees: the *global* GIB compresses *all* non-target variables, whereas the *Markov-blanket* GIB (MB–GIB) restricts the encoder to $\mathrm{MB}(T)$ and so never encodes non-blanket directions in the first place. Results are reported in Table 1 and Fig. 2.

Table 1: Average error metrics under spurious-proxy shift (MB-invariance holds) for the motivating example.

| Method | MAE | RMSE | $R^2$ |
|---|---|---|---|
| Markov Blanket GIB | **0.82** | **1.02** | **0.82** |
| Global GIB | 7.96 | 10.03 | -15.89 |

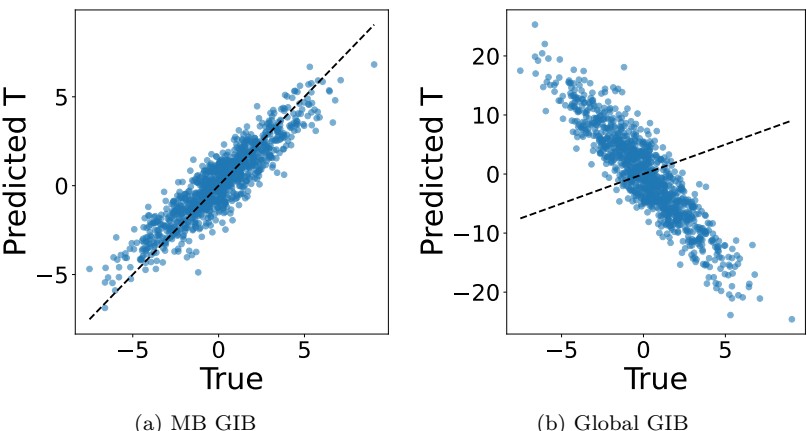

(a) MB GIB      (b) Global GIB

Figure 2: True vs. predicted (imputed) $T$ under the spurious-proxy shift. The Markov-blanket GIB remains accurate because it encodes only $C = \mathrm{MB}(T)$, for which $p_s(T \mid C) = p_t(T \mid C)$ holds exactly. In contrast, the global GIB collapses: the high-dimensional proxy block $S$ dominates the bottleneck in the source, but its relationship to $T$ reverses in the target (sign flip), yielding anti-predictive representations and large negative $R^2$.

**Discussion.** This example highlights a sharp, theory-aligned separation between global and Markov-blanket scopes. Because the target mechanism is invariant, restricting the encoder to $\mathrm{MB}(T) = \{C_1, \ldots, C_k\}$ preserves precisely the information required for transfer, and the MB GIB achieves strong target performance ($R^2 \approx 0.82$). By contrast, the global encoder has access to a large set of non-blanket variables $S$ that are highly predictive in the source yet shift substantially across domains; under a tight bottleneck, the global

GIB preferentially compresses these dominant proxy directions, leading to systematic failure in the target (RMSE 10.03, $R^2 = -15.89$). Taken together, the results demonstrate that a *DAG-aware Information Bottleneck* with a Markov-blanket encoder can provide a principled and practically robust solution to domain adaptation, precisely in the regime where the MB-invariance guarantee applies.

We summarize our main contributions as follows. We emphasize at the outset that predicting from $\mathrm{MB}(T)$ under blanket invariance is, by itself, a consequence of standard Markov-blanket sufficiency; our contributions are the specific estimators built on top of this restriction and their analysis, not the restriction principle itself.

- **Blanket restriction as the transfer mechanism (Section 3).** We make explicit that the structural restriction to $\mathrm{MB}(T)$—not the choice of estimator—is what drives zero-shot transfer under blanket invariance, and we delineate the setting (local causal knowledge, stable target mechanism) in which this is justified.

- **Closed-form MB–GIB with a lossless-restriction guarantee (Section 4.1).** For the linear–Gaussian case we give a closed-form, blanket-restricted Gaussian IB solution that reduces to CCA, and we prove it is *lossless* relative to using all non-target variables for any bottleneck dimension.

- **Practical MB–VIB for nonlinear/non-Gaussian regimes (Section 4.2).** Where no closed-form conditional estimator exists, we instantiate a variational, blanket-restricted IB with flexible decoders that trains on a single source environment and deploys zero-shot. Here the IB objective contributes principled compression and regularization, which is its primary benefit; in well-specified linear–Gaussian settings a plain blanket regression is already near-optimal.

- **Supporting theory (Sections 4.3 and 4.4).** We give population identifiability and zero-shot risk-preservation results under blanket invariance, and estimator-specific finite-sample guarantees for MB–GIB. These finite-sample results are framed as supportive guarantees for the proposed estimator rather than as the primary source of novelty.

- **Empirics (Section 5).** On synthetic and real datasets, blanket-restricted bottlenecks achieve accurate imputations under severe shifts and substantially outperform unrestricted baselines; consistent with theory, gains of IB over plain blanket regression appear in nonlinear/non-Gaussian regimes rather than in the well-specified linear–Gaussian one.

## 2 Related Work

**Origins and Early Development of the Information Bottleneck**  The Information Bottleneck (IB) was introduced by Tishby, Pereira, and Bialek (1999) (Tishby et al., 2000) as a principled way to extract task-relevant structure from data: learn a stochastic code (T) of (X) that maximizes mutual information with a target (Y) while compressing information about (X). Early work established the Lagrangian trade-off $I(X;T) - \beta I(T;Y)$, the self-consistent IB equations, and the "information curve" characterizing relevance vs. compression. Soon after, efficient clustering algorithms appeared—most notably the Agglomerative IB (AIB) of Slonim and Tishby—which applied IB to unsupervised/text settings and gave scalable heuristics for discrete variables (Slonim & Tishby, 1999; Slonim et al., 2001). On the theory side, Gaussian IB (Chechik et al., 2003) provided closed-form solutions for jointly Gaussian variables and linked IB to canonical correlation analysis and rate–distortion theory. Variants such as multi-variate IB (Slonim et al., 2006) and deterministic annealing schemes broadened applicability, and by the mid-2000s IB had become a unifying lens for representation learning, lossy compression with task-aware distortions, and information-theoretic clustering—setting the stage for later variational/deep renditions.

**From Theory to Deep Learning: The Variational Turn of IB**  The next wave of IB research made the framework practical for high-dimensional data by introducing variational surrogates and neural parameterizations. The *Variational Information Bottleneck* casts IB as a stochastic encoder–decoder trained with the reparameterization trick, optimizing a predictive loss plus a KL penalty to a simple prior; the trade-off coefficient $\beta$ becomes a tunable "information budget" (Alemi et al., 2016). This bridged IB with modern generative modeling—most visibly $\beta$-VAE as an unsupervised, rate–distortion analogue (Burgess et al.,

2018)—and enabled end-to-end training on images, text, and speech. In parallel, deterministic and annealed variants (e.g., the Deterministic Information Bottleneck) and "information plane" studies explored how compression emerges during deep training, sparking debate about generalization and the role of noise (Strouse & Schwab, 2017; Tishby & Zaslavsky, 2015). Subsequent work broadened IB's scope: nonlinear/non-Gaussian decoders and tighter variational bounds on mutual information (Poole et al., 2019), connections to dropout and noise injection as implicit bottlenecks (Achille & Soatto, 2018), and task-aware objectives such as the Conditional Entropy Bottleneck for robust prediction (Fischer, 2020).

**IB and Causal Inference.** Recent work integrates the Information Bottleneck with causal goals to obtain representations that are stable under interventions and distribution shift. A formal "causal IB" reframes sufficiency in interventional terms, proposing an objective that *compresses $X$* while *preserving causal control* of a target $Y$—yielding a CIB Lagrangian and an axiomatic notion of optimal causal variable abstractions that can be reasoned about with backdoor adjustment and without full DAG knowledge (Simoes et al., 2025). Beyond interventional sufficiency, causal IB ideas have been used to *separate robust (causal) from non-robust (spurious/style)* features via instrumental-variable style interventions, improving adversarial robustness by down-weighting non-causal signal within the bottleneck (Hua et al., 2022). For domain generalization, *Invariant Information Bottleneck* (IIB) casts invariant causal prediction in mutual-information terms and regularizes $I(Z; X)$ to avoid pseudo-invariant and geometrically skewed shortcuts, delivering tractable variational losses and consistent OOD gains (Li et al., 2022). Together, these strands position IB as a bridge between predictive compression and causal transportability: learn minimal, interventionally sufficient codes that transfer across environments while suppressing spurious variability.

**How our MB–GIB and MB–VIB differ.** Our approach is *DAG-aware and node-specific*: we place the bottleneck downstream of the *Markov blanket of the target $T$* and learn summaries solely from $\mathrm{MB}(T)$ (or parents) to impute/predict $T$ under shift. In the linear–Gaussian case, MB–GIB is a *closed-form* IB/CCA solution with an explicit blanket constraint; MB–VIB is its *nonlinear, non-Gaussian* variational counterpart (stochastic encoder; Student-$t$/Laplace heads). This differs from *Causal IB* (Simoes et al., 2025), which optimizes an interventional sufficiency objective to produce *causal abstractions* (possibly without full DAG knowledge) aimed at preserving *causal control* rather than enforcing a fixed blanket restriction. It also differs from adversarial-robustness uses of CIB (Hua et al., 2022), which introduce IV-style interventions/adversarial criteria to separate robust from non-robust features, whereas we exploit *known causal structure* to preclude non-blanket pathways a priori. Finally, unlike *Invariant Information Bottleneck* (Li et al., 2022), which regularizes mutual information to induce environment-level invariance (often requiring multiple environments or environment labels), our invariance comes from *structural* restriction: by confining the encoder to $\mathrm{MB}(T)$, we keep mechanism-stable, $T$-relevant signals and discard context- or downstream-induced variability. Practically, this yields a simple pipeline: (i) graph-constrained encoder (MB or parents), (ii) closed-form GIB or variational VIB training on a *single* source environment, and (iii) zero-shot deployment to shifted targets, without interventional data, adversarial training, or multi-environment supervision.

**When MB–GIB / MB–VIB are most useful.** Our DAG-aware bottlenecks are most powerful when a (partial) graph around the target $T$ is known, training comes from a *single* source environment, and deployment faces *contextual* shifts while mechanisms are stable. They are especially effective when $T$ is unobserved at test time, data are limited (closed-form MB–GIB is sample-efficient), or robustness is critical: by restricting the encoder to $\mathrm{MB}(T)$, they transfer zero-shot, avoid spurious downstream paths, and require neither interventional data nor multi-environment supervision.

## 3 Problem Statement

We study *domain adaptation with local causal knowledge*: we impute a designated *target variable $T$* across environments when only *local causal information* around $T$ (its parents or Markov blanket) is available and the data may be non-Gaussian. This is a deliberately scoped setting rather than fully unsupervised domain adaptation. It is practically meaningful whenever partial structural knowledge around $T$ is available—for example, from domain expertise, an established mechanistic model, or local causal discovery run on the source—and the mechanism $p(T \mid \mathrm{MB}(T))$ is believed stable across environments (e.g., a fixed biological or

physical law observed under different experimental or sampling conditions). We make this scope explicit so that our contribution is read as a closed-form/variational estimator that exploits known local structure, not as a method that discovers invariance from scratch.

**Notation: variables and environments.** We index *environments* by $e \in \{s, t\}$ (source and target) and reserve roman subscripts $s, t$ for environments throughout. The target $T$ is a designated variable; when its position must be named explicitly we write $T = X_\tau$ with node index $\tau$, so the environment label and the node index are never conflated.

**Data and observables.** Let $X = (X_1, \ldots, X_p)$ be $p$ observed variables. We observe a *source* dataset with $T$ present,

$$\mathcal{D}_s = \{X_s^{(i)}\}_{i=1}^{N_s}, \qquad X_s^{(i)} = (X_{1,s}^{(i)}, \ldots, X_{p,s}^{(i)}),$$

and a *target* dataset in which $T$ is entirely missing,

$$\mathcal{D}_t^{\text{obs}} = \{X_{-\tau,t}^{(j)}\}_{j=1}^{N_t}, \qquad X_{-\tau,t}^{(j)} = (X_{1,t}^{(j)}, \ldots, X_{\tau-1,t}^{(j)}, X_{\tau+1,t}^{(j)}, \ldots, X_{p,t}^{(j)}).$$

**Graphical knowledge: Markov blanket of $T$.** We *do not assume* the full DAG is known. Instead, we assume access to (or can reliably estimate) the *Markov blanket* of $T$ in some underlying DAG $\mathcal{G}$:

$$\text{MB}(T) = \text{Pa}(T) \cup \text{Ch}(T) \cup \bigcup_{C \in \text{Ch}(T)} \text{Pa}(C) \setminus \{T\},$$

i.e., the union of $T$'s parents, its children, and the parents of those children (spouses). By the local Markov property, $T \perp X \setminus (\text{MB}(T) \cup \{T\}) \mid \text{MB}(T)$. In words, conditioning on $\text{MB}(T)$ renders $T$ independent of all remaining nodes.

**Model class (beyond Gaussian).** We allow *arbitrary* (possibly nonlinear, non-Gaussian) structural mechanisms and noise distributions; no linearity or Gaussianity is required. Throughout, we make no parametric restrictions on the joint distribution other than the blanket conditional independence above.

**Shift model (MB invariance).** Source and target environments may differ (covariate/target shifts) in the marginals of exogenous or downstream variables and in joint correlations outside the blanket. Our sole stability assumption is *MB invariance*:

$$p_s(T \mid \text{MB}(T)) = p_t(T \mid \text{MB}(T)),$$

i.e., the conditional law of $T$ given its blanket is unchanged across domains. Under this assumption, shifts outside $\text{MB}(T)$ cannot alter $p(T \mid \text{MB}(T))$ by conditional independence; shifts *within* $\text{MB}(T)$ may occur in the marginals of blanket variables but not in the mechanism linking $T$ to its blanket.

**Task.** Let $M := \text{MB}(T)$ be the index set of blanket variables, and write $X_M = (X_k)_{k \in M}$. For the $j$-th target sample, denote its blanket subvector by $x_{M,t}^{(j)} := (X_{k,t}^{(j)})_{k \in M}$. Given $(\mathcal{D}_s, \mathcal{D}_t^{\text{obs}}, M)$ and the MB-invariance assumption $p_s(T \mid X_M) = p_t(T \mid X_M)$, we impute the missing targets via the (source-trained) Bayes predictor

$$\widehat{T}^{(j)} = \mathbb{E}_{p_s}[T \mid X_M = x_{M,t}^{(j)}], \qquad j = 1, \ldots, N_t.$$

The equality of conditionals ensures *zero-shot transfer* from source to target. Practically, we estimate this conditional with a *DAG-aware Information Bottleneck* restricted to the blanket inputs $X_M$ (or, when downstream paths are deemed unstable, to the parents-only scope $X_P$ with $P := \text{Pa}(T) \subseteq M$): a closed-form, sample-efficient linear MB–GIB, and a nonlinear MB–VIB (stochastic encoder/decoder) for non-Gaussian, nonlinear settings. Unless stated otherwise, the encoder scope is the full Markov blanket $X_M$; we flag explicitly wherever the parents-only scope $X_P$ is used instead.

## 4 Methodology and Theoretical Results

We develop a *DAG-aware Information Bottleneck* pipeline tailored to imputing an unobserved target $T$ under domain shift using only local causal knowledge. Our methodology enforces a structural input constraint—encoders see either the parents $\mathrm{Pa}(T)$ or the Markov blanket $\mathrm{MB}(T)$—and exploits the *MB-invariance* assumption $p_{\mathrm{s}}(T \,|\, X_{\mathrm{MB}}) = p_{\mathrm{t}}(T \,|\, X_{\mathrm{MB}})$ to enable zero-shot transfer. Concretely, we present (i) a closed-form **MB–GIB** for the linear–Gaussian case, where the GIB subspace equals CCA directions and we prove that restricting to $\mathrm{MB}(T)$ is *lossless* relative to using all non-$T$ variables; and (ii) a **MB–VIB** with stochastic encoders and flexible (Laplace/Student-$t$) decoders for nonlinear, non-Gaussian settings, trained by a predictive loss plus a KL compression term. On the theory side, we show (a) equivalence of global and MB-restricted GIB spectra in the Gaussian case and a block-matrix proof that all optimal directions lie in the lifted $\mathrm{MB}(T)$ subspace; (b) identifiability of the population predictor $\mathbb{E}[T \,|\, X_{\mathrm{MB}}]$ from source data and its risk preservation in the target domain under MB-invariance; and (c) finite-sample guarantees that the MB–GIB estimator concentrates around the population conditional. We stress the division of labor: the *transfer* guarantee comes from the combination of blanket restriction and MB-invariance ((a)–(b)), while the finite-sample results in (c) are *estimator-specific* guarantees that quantify how well the source estimator recovers that transferable predictor; they are supportive theory for MB–GIB rather than DA-specific results. Together, these results justify using $\mathrm{MB}(T)$ as a *structurally minimal* and *transfer-stable* interface for bottleneck learning, and they motivate our practical algorithms that remain sample-efficient (MB–GIB) yet expressive (MB–VIB) across a wide range of data regimes.

**What does the IB objective add beyond blanket regression?** Because the encoder already sees only $X_M$, one may ask whether the IB objective contributes anything over directly regressing $T$ on $X_M$. We are deliberately careful here. In the well-specified linear–Gaussian regime, the population predictor is $g^{\star}(x_M) = \mathbb{E}[T \mid X_M = x_M]$ and ordinary least squares on $X_M$ already attains it; MB–GIB, which reduces to CCA in this regime (Sect. 4.1), provides no accuracy gain over a plain linear baseline, and an over-aggressive bottleneck can even *over-compress*. The value of the IB formulation is therefore twofold and orthogonal to accuracy in the linear–Gaussian case: (i) it supplies a principled compression/regularization knob ($\beta$) that controls $I(X_M; Z)$ and suppresses nuisance variation, useful at small samples; and (ii) it extends naturally to nonlinear and non-Gaussian regimes via MB–VIB, where no closed-form conditional estimator is available and flexible encoders/decoders are needed. We make these claims falsifiable in Section 5 by comparing against tuned blanket-restricted regressors.

**Two kinds of shift.** Throughout, we distinguish two qualitatively different regimes whenever we claim robustness. *Shifts outside* $\mathrm{MB}(T)$ (the marginals of, or dependencies among, non-blanket variables) cannot change $p(T \mid X_M)$ by conditional independence, and are exactly the regime in which our zero-shot guarantees hold. *Drift of the blanket mechanism $p(T \mid X_M)$ itself* (e.g., a new parent of $T$ or an altered noise law) violates MB-invariance and breaks the guarantees; Corollary 2 quantifies the resulting degradation, and Section 5.4 discusses diagnostics.

### 4.1 MB–GIB: Closed-Form Solution and Lossless Restriction (Gaussian)

In the linear–Gaussian case, whitening $X_M$ and $T$ turns the IB tradeoff into finding directions in $X_M$ that are *maximally correlated* with $T$. This is exactly canonical correlation analysis (CCA): solve a symmetric eigenproblem for the operator $\Omega_{X_M} = \Sigma_{X_M X_M}^{-1/2} \Sigma_{X_M T} \Sigma_{TT}^{-1} \Sigma_{T X_M} \Sigma_{X_M X_M}^{-1/2}$, take its top $d$ eigenvectors, un-whiten to get $W$, and encode $Z = W^{\top} X_M$. Because the model is Gaussian and linear, these $Z$ are *sufficient statistics* for predicting $T$: no other linear combination of $X_M$ carries additional predictive information once $Z$ is known. The decoder is just the optimal linear predictor of $T$ from $Z$ (ordinary least squares): $\hat{T} = B^{\top} Z$ with $B = (Z^{\top} Z)^{-1} Z^{\top} T$. The IB parameter $\beta$ acts as a *spectral threshold*: only CCA directions with squared canonical correlation above a $\beta$-dependent cutoff are kept (equivalently, choose $d$ to retain the leading spectrum). Under the Markov-blanket restriction $X_M = \mathrm{MB}(T)$, the nonzero spectrum matches the global one, so this closed-form encoder/decoder is *lossless* relative to using all non-$T$ variables.

**Theorem 1** (Lossless blanket restriction, Gaussian/linear)**.** *Let $X = (X_{-\tau}, T)$ be jointly Gaussian and $M = \mathrm{MB}(T)$. Assuming $T \perp X \setminus (M \cup \{T\}) \mid M$, the nonzero spectra of the global CCA operator $\Omega_X$ and*

*the blanket operator* $\Omega_M$ *coincide, and every global CCA direction lies in the lifted subspace generated by $M$. Hence restricting the encoder to* $\mathrm{MB}(T)$ *is without loss for any bottleneck dimension d, and the Bayes risk of predicting $T$ from $X$ equals that from $M$.*

The full statement and proof, via the Markov-blanket factorization $\Sigma_{XT} = L\,\Sigma_{MT}$ with $L = [I; B]$ and the metric identity $L^\top \Sigma_{XX}^{-1} L = \Sigma_{MM}^{-1}$ (which yields $S_X = Q\,S_M$ with $Q$ column-orthonormal, so $S_X$ and $S_M$ share singular values), are given in Appendix A (Theorem 5).

**Practical recipe and cost.** Compute empirical covariances on source $(X_M, T)$, form $\Omega_{X_M}$, take its top $d$ eigenvectors to obtain $W$, and fit $B$ by least squares from $Z$ to $T$. Complexity is $O(n|X_M|^2 + |X_M|^3)$, typically small since $|X_M| = |M|$. At test time, encode $z = W^\top x_{M,\mathrm{t}}$ and predict $\hat{T} = B^\top z$.

## 4.2 MB–VIB: Nonlinear, Non-Gaussian Bottleneck (Practical Variant)

When relationships are nonlinear or noises are non-Gaussian, we use a variational bottleneck restricted to the blanket inputs $X_M$. The encoder is a stochastic map $q_\phi(z \mid x_M)$ (neural mean/variance with reparameterization), the decoder a flexible likelihood $q_\theta(t \mid z)$ (Gaussian/Laplace/Student-$t$ as appropriate). We train on source pairs $(x_M, t)$ by minimizing

$$\mathcal{L}_{\mathrm{VIB}}(\phi, \theta) = \mathbb{E}_{(x_M, t)}\,\mathbb{E}_{z \sim q_\phi(\cdot \mid x_M)}\big[-\log q_\theta(t \mid z)\big]$$
$$+\ \beta\,\mathbb{E}_{x_M}\,D_{\mathrm{KL}}\big(q_\phi(z \mid x_M)\,\|\,r(z)\big).$$

with a simple prior $r(z)$ (e.g., $\mathcal{N}(0, I)$). The first term fits a predictive decoder for $T$; the KL controls information $I(X_M; Z)$ and thus shrinks nuisance variability that does not help predict $T$. The DAG-aware input restriction prevents leakage from non-blanket pathways, aligning the learned representation with the transfer-stable mechanism $p(T \mid X_M)$. The population-level justification for restricting the variational bottleneck to $X_M$—namely that $T \perp X \setminus (M \cup \{T\}) \mid M$ implies $I(T; X) = I(T; M)$ and $p(T \mid X) = p(T \mid M)$, so a blanket-only encoder weakly dominates in the IB objective—is distribution-free and does not require Gaussianity; see Appendix A (Proposition 1).

*Design choices.* (i) Likelihood: choose $q_\theta$ to match the target type (e.g., Student-$t$ for heavy tails); (ii) Capacity: adjust $z$-dim and $\beta$ to trade accuracy for robustness; (iii) Stability: standardization, early stopping, and prior tempering improve optimization. *Deployment* is simple: encode $z = f_\phi(x_{M,\mathrm{t}})$ from target inputs restricted to $X_M$, then output $\hat{T} = \mathbb{E}[T \mid z]$ (regression) or $\arg\max_y q_\theta(y \mid z)$ (classification). The blanket restriction prevents leakage from non-blanket pathways, while the variational bottleneck captures nonlinear predictors that remain stable under shifts outside $\mathrm{MB}(T)$.

## 4.3 Transfer Guarantees under MB Invariance

We give population-level transfer guarantees for blanket-restricted IB predictors; the finite-sample counterparts and full proofs appear in Section 4.4 and Appendix B.

**Identifiability (population).** Let $M = \mathrm{MB}(T)$ and assume $p_{\mathrm{s}}(T \mid X_M) = p_{\mathrm{t}}(T \mid X_M)$. Then the Bayes rule

$$g^\star(x_M)\ =\ \mathbb{E}_{p_{\mathrm{s}}}[T \mid X_M = x_M]\ =\ \mathbb{E}_{p_{\mathrm{t}}}[T \mid X_M = x_M]$$

is identifiable from source data alone and is target-optimal conditional on $X_M$.

**Risk preservation (zero-shot).** For squared or log loss, any estimator $\hat{g}$ trained on source pairs $(X_M, T)$ and applied to target inputs satisfies

$$\mathcal{R}_{\mathrm{t}}(\hat{g}) - \mathcal{R}_{\mathrm{t}}(g^\star)\ =\ \mathcal{R}_{\mathrm{s}}(\hat{g}) - \mathcal{R}_{\mathrm{s}}(g^\star),$$

so source excess risk transfers verbatim to the target. In particular: (i) *MB–GIB losslessness* in the Gaussian/linear case (Sec. 4.1) shows restricting inputs to $X_M$ is without loss for any bottleneck dimension; (ii) *MB–VIB consistency* (Sec. 4.2) implies target consistency whenever source risk vanishes; and (iii) by $T \perp X \setminus (M \cup \{T\}) \mid X_M$, shifts outside $M$ cannot change the optimal predictor.

**Practical Considerations** (1) *Checking MB invariance.* Compare source vs. target residuals or decoder log-likelihoods of $\hat{g}(X_{M,\mathrm{t}})$; systematic shifts flag drift in $p(T \mid X_M)$. (2) *Estimating M when unknown.* When the blanket is not supplied by domain knowledge, we estimate it on the source with local causal discovery around $T$: recover a candidate parent/child/spouse set (e.g., via a grow–shrink or HITON-style local search, or by thresholding a learned local DAG), then prune with conditional-independence tests, retaining a variable $X_k$ in $\widehat{M}$ only if $T \not\perp X_k \mid \widehat{M} \setminus \{X_k\}$. We validate the estimate on the source by goodness-of-fit of $p_{\mathrm{s}}(T \mid X_{\widehat{M}})$ and by checking residual stability between a source hold-out and the target. This estimation step has well-known failure modes that can break the blanket conditional independence or the invariance assumption: *hidden confounding* (an unobserved common cause of $T$ and some $X_k$), *finite-sample instability* of CI tests in high dimensions, *measurement error* in blanket variables, and *model misspecification* of the CI test or score. Section 5.4 and the misspecification study in Section 5 (Table 2) quantify the empirical cost of an imperfectly recovered blanket, and we recommend the parents-only fallback and increased compression when these diagnostics fire. (3) *Tuning.* Increase $\beta$ and decrease $d_z$ for robustness; select decoder likelihood to match tails (Gaussian/Laplace/Student-$t$).

## 4.4 Finite-Sample Guarantees and Robustness

We now make the finite-sample and robustness claims in Secs. 4.1–4.3 explicit. The goal is twofold: (i) quantify how the *MB–GIB* estimator (which is computed from empirical covariances) concentrates around its population solution (equivalently, around the population conditional $g^\star(x_M) = \mathbb{E}[T \mid X_M = x_M]$); and (ii) formalize robustness under distribution shift, clarifying which guarantees require exact MB invariance and which extend to small violations.

**Setup and notation.** Let $M = \mathrm{MB}(T)$ and denote $X_M \in \mathbb{R}^p$ with $p := |M|$. We observe i.i.d. *source* samples $\{(X_{M,i}, T_i)\}_{i=1}^n \sim p_{\mathrm{s}}(X_M, T)$. Let $Z := (X_M^\top, T)^\top \in \mathbb{R}^{p+1}$ with population covariance

$$\Sigma := \mathrm{Cov}(Z) = \begin{pmatrix} \Sigma_{XX} & \Sigma_{XT} \\ \Sigma_{TX} & \Sigma_{TT} \end{pmatrix}, \qquad \widehat{\Sigma} := \mathrm{Cov}_n(Z)$$

the corresponding sample covariance (with delta degrees of freedom $ddof = 1$). In the Gaussian/linear regime of Sec. 4.1, the MB–GIB directions are the top eigenvectors of the CCA operator

$$\Omega_M := \Sigma_{XX}^{-1/2} \Sigma_{XT} \Sigma_{TT}^{-1} \Sigma_{TX} \Sigma_{XX}^{-1/2} \in \mathbb{R}^{p \times p},$$

and the empirical estimator replaces $\Sigma$ by $\widehat{\Sigma}$, yielding $\widehat{\Omega}_M$. Let $U_\star \in \mathbb{R}^{p \times d}$ collect the top $d$ eigenvectors of $\Omega_M$, and $\widehat{U}$ those of $\widehat{\Omega}_M$. We measure subspace error using the principal-angle quantity $\|\sin\Theta(\widehat{U}, U_\star)\|_{\mathrm{op}}$, and write $\Pi_\star = U_\star U_\star^\top$, $\widehat{\Pi} = \widehat{U}\widehat{U}^\top$ for the projectors.

**Assumptions (finite-sample MB–GIB).** The following conditions are standard in non-asymptotic covariance and spectral analysis, and they are satisfied by the linear–Gaussian SEMs used in our experiments.

**Assumption 1** (Joint sub-Gaussian source law). *Let $Z := (X_M^\top, T)^\top \in \mathbb{R}^{p+1}$ denote the source random vector with $\mathbb{E}(Z) = 0$ and covariance $\Sigma$. We assume $Z$ is $K$-sub-Gaussian in the standard (joint) sense:*

$$\|Z\|_{\psi_2} := \sup_{\|u\|_2=1} \|u^\top Z\|_{\psi_2} \leq K,$$

*where $\|\cdot\|_{\psi_2}$ is the sub-Gaussian Orlicz norm. (In particular, the linear–Gaussian SEM case satisfies this condition.)*

**Assumption 2** (Blanket covariance conditioning). *The blanket covariance is well-conditioned: $\lambda_{\min}(\Sigma_{X_M X_M}) \geq \lambda_0 > 0$ and $\lambda_{\max}(\Sigma_{X_M X_M}) \leq \Lambda_0 < \infty$.*

**Assumption 3** (Spectral gap for the retained CCA spectrum). *Let $\lambda_1 \geq \cdots \geq \lambda_p \geq 0$ be the eigenvalues of $\Omega_M$. For the chosen bottleneck dimension $d$, there is an eigengap $\gamma := \lambda_d - \lambda_{d+1} > 0$ (with $\lambda_{p+1} := 0$).*

**(A) Concentration of empirical covariances.** Theorem 2 quantifies estimation error of the covariance blocks used by MB–GIB.

**Theorem 2** (Covariance concentration). *Under Assumption 1, for any $\delta \in (0,1)$, with probability at least $1 - \delta$,*

$$\|\widehat{\Sigma} - \Sigma\|_{\mathrm{op}} \leq c\,K^2 \left( \sqrt{\frac{p + 1 + \log(1/\delta)}{n}} + \frac{p + 1 + \log(1/\delta)}{n} \right),$$

*where $c > 0$ is a universal constant. In particular, the same bound holds for each block $\widehat{\Sigma}_{XX}, \widehat{\Sigma}_{XT}, \widehat{\Sigma}_{TT}$.*

**(B) Finite-sample stability of the MB–GIB spectrum.** Theorem 3 translates covariance error into *spectral* (subspace) error for the CCA/GIB directions, with the eigengap $\gamma$ controlling sensitivity.

**Theorem 3** (MB–GIB spectral/subspace concentration). *Assume Assumptions 1–3. Let $U_\star$ and $\widehat{U}$ be the population and empirical top-$d$ eigenspaces of $\Omega_M$ and $\widehat{\Omega}_M$, respectively. Then, with probability at least $1 - \delta$,*

$$\|\sin\Theta(\widehat{U}, U_\star)\|_{\mathrm{op}} \leq \frac{C}{\gamma} \|\widehat{\Omega}_M - \Omega_M\|_{\mathrm{op}} \leq \frac{C'}{\gamma} \|\widehat{\Sigma} - \Sigma\|_{\mathrm{op}} \lesssim \frac{K^2}{\gamma} \sqrt{\frac{p + \log(1/\delta)}{n}},$$

*where $C, C' > 0$ depend only on the conditioning constants $\lambda_0, \Lambda_0$ (Assumption 2). Equivalently, $\|\widehat{\Pi} - \Pi_\star\|_{\mathrm{op}} \leq 2\|\sin\Theta(\widehat{U}, U_\star)\|_{\mathrm{op}}$.*

**(C) Finite-sample excess risk (source) and transfer (target).** Theorem 4 yields an explicit $O\big((p + \log(1/\delta))/n\big)$ excess-risk rate that depends only on the blanket dimension, formalizing the sample-efficiency benefit of restricting inputs to MB($T$).

**Theorem 4** (MB–GIB excess risk rate). *Assume Assumptions 1–2. Let $g^\star(x_M) = \mathbb{E}_{p_{\mathrm{s}}}[T \mid X_M = x_M]$ denote the population conditional mean. Let $\widehat{g}$ be the MB–GIB predictor obtained by (i) computing $\widehat{\Omega}_M$, (ii) taking $\widehat{U}$ (or equivalently the corresponding unwhitened $W$), and (iii) fitting the optimal linear decoder from $Z = \widehat{W}^\top X_M$ to $T$ (Sec. 4.1). Then, for squared loss $\mathcal{R}_{\mathrm{s}}(g) := \mathbb{E}_{p_{\mathrm{s}}}\big[(T - g(X_M))^2\big]$, with probability at least $1 - \delta$,*

$$\mathcal{R}_{\mathrm{s}}(\widehat{g}) - \mathcal{R}_{\mathrm{s}}(g^\star) \leq C'' \|\widehat{\Sigma} - \Sigma\|_{\mathrm{op}}^2 \lesssim K^4 \frac{p + \log(1/\delta)}{n},$$

*where $C'' > 0$ depends only on $\lambda_0, \Lambda_0$.*

Corollaries 1–2 formalize robustness under blanket-based transfer. Under exact MB invariance, the Bayes rule $g^\star(x_M) = \mathbb{E}[T \mid X_M = x_M]$ is shared across domains, and the target excess risk is controlled by the source excess risk up to a blanket density-ratio factor $\rho_M$; in the special case $p_{\mathrm{s}}(X_M) = p_{\mathrm{t}}(X_M)$ this reduces to an exact equality. Under approximate invariance, departures from the ideal transfer regime are quantified by the conditional-mean mismatch $\Delta(x_M)$ within the blanket, yielding an explicit degradation bound in terms of $\mathbb{E}_{p_t}|\Delta|$ and $\mathbb{E}_{p_t}\Delta^2$. Finally, for the nonlinear MB–VIB variant, analogous non-asymptotic guarantees require explicit capacity control for the encoder/decoder (e.g., Lipschitz/covering-number conditions or PAC-Bayes complexity). Our main theoretical guarantees therefore target MB–GIB (Gaussian/linear), while MB–VIB is a practical extension motivated by the same structural restriction and validated empirically.

**Corollary 1** (Zero-shot target bound under MB invariance). *Assume MB invariance $p_{\mathrm{s}}(T \mid X_M) = p_{\mathrm{t}}(T \mid X_M)$, squared loss, and that $p_{\mathrm{t}}(X_M) \ll p_{\mathrm{s}}(X_M)$ with*

$$\rho_M := \operatorname*{ess\,sup}_{x_M} \frac{p_{\mathrm{t}}(x_M)}{p_{\mathrm{s}}(x_M)} < \infty.$$

*Then the Bayes predictor*

$$g^\star(x_M) = \mathbb{E}[T \mid X_M = x_M]$$

*is shared across domains and is target-optimal among all predictors measurable w.r.t. $X_M$. Moreover, for any estimator $\widehat{g}$ (in particular, MB–GIB) we have the conditional excess-risk identity*

$$\mathcal{R}_\nu(\widehat{g}) - \mathcal{R}_\nu(g^\star) = \mathbb{E}_{p_\nu(X_M)}\big[(\widehat{g}(X_M) - g^\star(X_M))^2\big], \qquad \nu \in \{\mathrm{s}, \mathrm{t}\}.$$

*Consequently,*

$$\mathcal{R}_{\mathrm{t}}(\widehat{g}) - \mathcal{R}_{\mathrm{t}}(g^{\star}) \;\leq\; \rho_M \left( \mathcal{R}_{\mathrm{s}}(\widehat{g}) - \mathcal{R}_{\mathrm{s}}(g^{\star}) \right).$$

*In particular, under the assumptions of Theorem 4, with probability at least $1 - \delta$,*

$$\mathcal{R}_{\mathrm{t}}(\widehat{g}) - \mathcal{R}_{\mathrm{t}}(g^{\star}) \;\lesssim\; \rho_M \, K^4 \, \frac{p + \log(1/\delta)}{n}.$$

*In the special case $p_{\mathrm{s}}(X_M) = p_{\mathrm{t}}(X_M)$, we have $\rho_M = 1$ and the target and source excess risks coincide.*

**Corollary 2** (Approximate MB invariance). *Assume squared loss and define the conditional-mean mismatch within the blanket by*

$$\Delta(x_M) \;:=\; \mathbb{E}_{p_{\mathrm{t}}}[T \mid X_M = x_M] - \mathbb{E}_{p_{\mathrm{s}}}[T \mid X_M = x_M] \;=\; g_{\mathrm{t}}^{\star}(x_M) - g_{\mathrm{s}}^{\star}(x_M).$$

*Then for any predictor $\widehat{g}$,*

$$\left| \left( \mathcal{R}_{\mathrm{t}}(\widehat{g}) - \mathcal{R}_{\mathrm{t}}(g_{\mathrm{t}}^{\star}) \right) - \mathbb{E}_{p_{\mathrm{t}}(X_M)}\big[ (\widehat{g}(X_M) - g_{\mathrm{s}}^{\star}(X_M))^2 \big] \right| \;\leq\; 2\,\mathbb{E}_{p_{\mathrm{t}}}[|\widehat{g}(X_M) - g_{\mathrm{s}}^{\star}(X_M)|\,|\Delta(X_M)|] + \mathbb{E}_{p_{\mathrm{t}}}\big[ \Delta(X_M)^2 \big].$$

*In particular, if shifts occur only outside $M$ then $\Delta \equiv 0$ and*

$$\mathcal{R}_{\mathrm{t}}(\widehat{g}) - \mathcal{R}_{\mathrm{t}}(g_{\mathrm{t}}^{\star}) \;=\; \mathbb{E}_{p_{\mathrm{t}}(X_M)}\big[ (\widehat{g}(X_M) - g_{\mathrm{s}}^{\star}(X_M))^2 \big],$$

*so the target excess risk is exactly the target $L_2$ error to the source conditional mean. More generally, target degradation is controlled by the blanket mismatch $\Delta$.*

## 4.5 Algorithms and Complexity

Here, we summarize training and deployment for MB–GIB and MB–VIB algorithms:

**MB–GIB (linear–Gaussian; closed form).** *Input:* source pairs $(X_M, T)$ (blanket-restricted inputs), bottleneck dim. $d$. *Steps:*

1. Standardize $X_M, T$ on the source; compute $\widehat{\Sigma}_{X_M X_M}, \widehat{\Sigma}_{X_M T}, \widehat{\Sigma}_{TT}$.

2. Form the CCA/IB operator $\widehat{\Omega}_{X_M} = \widehat{\Sigma}_{X_M X_M}^{-1/2} \widehat{\Sigma}_{X_M T} \widehat{\Sigma}_{TT}^{-1} \widehat{\Sigma}_{T X_M} \widehat{\Sigma}_{X_M X_M}^{-1/2}$.

3. Take top-$d$ eigenvectors $\widehat{W}$ of $\widehat{\Omega}_{X_M}$; encode $Z = \widehat{W}^{\top} X_M$.

4. Fit decoder by least squares: $\widehat{B} = \arg\min_B \|T - Z^{\top} B\|_2^2$; at test time predict $\hat{T} = \widehat{B}^{\top} \widehat{W}^{\top} x_{M,\mathrm{t}}$.

*Cost:* covariance $O(n|X_M|^2)$; eigen-decomp. $O(|X_M|^3)$; least squares $O(nd^2)$.

**MB–VIB (nonlinear, non-Gaussian; variational).** *Input:* source pairs $(X_M, T)$, encoder $q_\phi(z \mid x_M)$, decoder $q_\theta(t \mid z)$, prior $r(z)$, trade-off $\beta$, epochs $E$. *Steps:*

1. Initialize $\phi, \theta$; standardize $X_M$; choose $q_\theta$ (Gaussian/Laplace/Student-$t$) to match $T$.

2. For $e = 1, \ldots, E$: sample minibatches, draw $z = \mu_\phi(x_M) + \sigma_\phi(x_M) \odot \epsilon$ (reparameterization).

3. Minimize

$$\mathcal{L}_{\mathrm{VIB}} = \mathbb{E}[-\log q_\theta(T \mid Z)] + \beta \, \mathbb{E}\big[ D_{\mathrm{KL}}(q_\phi(Z \mid X_M) \,\|\, r(Z)) \big]$$

   by SGD; apply early stopping on source validation loss.

4. At test time, encode $z = f_\phi(x_{M,\mathrm{t}})$; output $\hat{T} = \mathbb{E}_{q_\theta}[T \mid z]$ (regression) or $\arg\max_y q_\theta(y \mid z)$ (classification).

*Cost:* per-epoch $O(n \cdot d_z \cdot H)$ where $d_z$ is latent dimension and $H$ is encoder/decoder width; memory linear in $n$ and model size.

**Notes.** (1) The *only* inputs to the encoder are the blanket variables $X_M$, enforcing the graphical constraint. (2) Choose $d$ or $d_z$ small and increase until validation loss saturates; increase $\beta$ for more robustness. (3) For heavy-tailed targets, prefer Student-$t$ decoders.

## 5 Experimental Results

In this section we evaluate DAG-aware Information Bottlenecks under domain shift. We report headline results on three settings: a controlled *7-node* SEM with known Markov blanket (for transparent diagnostics), the *64-node MAGIC–IRRI* Gaussian Bayesian network (benchmark scale; (Scutari, 2016)[1]), and a real single-cell signaling dataset from *Sachs et al.* (Sachs et al., 2005), which provides a stringent biological transfer test where the target (T) is unobserved at deployment. Across all datasets we compare **MB–GIB** and **MB–VIB** against baselines—Bayesian network (BN), pure deep neural network (DNN), and an IIB-style variant (Li et al., 2022)—and, on MAGIC–IRRI, against tuned blanket-restricted direct regressors (linear/ridge/kernel-ridge/MLP on MB($T$) with no IB objective) that isolate the contribution of the IB objective, under covariate and generalized target shifts, using MAE/RMSE/($R^2$) (mean($\pm$)SE over seeds) and runtime. To conserve space, the more exhaustive *ablations* (scope: Parents/MB/Global; capacity and ($\beta$); likelihood) and *sensitivity* analyses (shift magnitude/type, support mismatch, missingness, ($N_s$) curves) are conducted on the 7-node SEM and deferred to the supplement; because the correctness of the Markov blanket is central to the method, however, we surface a blanket-misspecification study directly in the main text (Table 2). The main text otherwise fixes hyperparameters from those sweeps and reports cross-dataset, zero-shot transfer performance. All experiments were executed on a Windows workstation equipped with a 12th Gen Intel(R) Core(TM) i9-12900H 2.50 GHz processor. The source code and experimental scripts for this work are available at the supplementary materials. Code to reproduce the experiments is available at `https://github.com/majavid/CDA_IB`.

### 5.1 Simulated Experiments: Controlled 7-Node SEM

First, we study a seven–node SEM where contexts $C_1, C_2$ generate intermediates $Z, X$, the treatment/latent target $T$ depends on $(C_1, X, Z)$, and $T$ drives downstreams $P, Y$. We adopt the *MB–invariance* setting: the Markov blanket MB($T$) = $\{C_1, X, Z, P, Y\}$ is shared across domains, while marginal distributions may shift. We consider two large–shift targets: (i) **covariate shift**, changing only $C_2$ (e.g., $C_2 \sim \mathcal{N}(5, 2^2)$) with all mechanisms and noise laws fixed; and (ii) **generalized target shift**, changing $C_1$ (e.g., $C_1 \sim \mathcal{N}(5, 2^2)$) *and* the disturbance/prior of $T$ via an additive offset and/or inflated $\varepsilon_T$ variance. In the target domain, $T$ is unobserved; we impute it zero–shot from observed variables using our DAG–aware bottlenecks, training the encoder/decoder on source pairs $(X_M, T)$ restricted to the blanket, and comparing MB–GIB/MB–VIB to BN, pure DNN, and IIB–style baselines. For the main results, we repeat each case 5 times.

Based on the results, depicted in Fig. 3, the **MB-GIB** model attains the strongest performance across metrics and scenarios, with the largest margin under generalized target shift. In both the covariate shift and target generation settings, **MB-GIB** achieves the lowest Root Mean Squared Error (RMSE) and Mean Absolute Error (MAE), alongside the highest $R^2$ score. This is expected rather than surprising: the data-generating process here is linear–Gaussian, so MB-GIB (which reduces to CCA, and whose linear decoder coincides with ordinary least squares on MB($T$)) is exactly matched to the model and recovers the optimal predictor without approximation error. We read this less as evidence that the IB objective is essential and more as a sanity check that the blanket-restricted estimator behaves as the theory predicts in the well-specified regime. The other models show more variable performance. The **BN** model performs well under the covariate shift but degrades markedly in the target generation task, where it exhibits the highest error and lowest $R^2$ score by a significant margin. Because the source-fit BN propagates the full source joint, it inherits source-specific dependencies that no longer hold once the target's own mechanism is perturbed, so its imputations break down under target-mechanism change even though it is competitive under pure covariate shift. The **MB-VIB**, **IIB-style**, and **PureDNN** models show intermediate performance: their general-purpose neural components introduce optimization and approximation error that a closed-form estimator avoids in this linear–Gaussian setting, so they trail MB-GIB here even though their flexibility is what pays

---

[1]The network structure and data are available at `https://www.bnlearn.com/bnrepository/`.

off in the nonlinear/non-Gaussian regimes studied below. This highlights that while several models handle simple covariate shifts, the closed-form blanket estimator is the natural choice when the structure is well-specified, precisely because it matches the problem exactly.

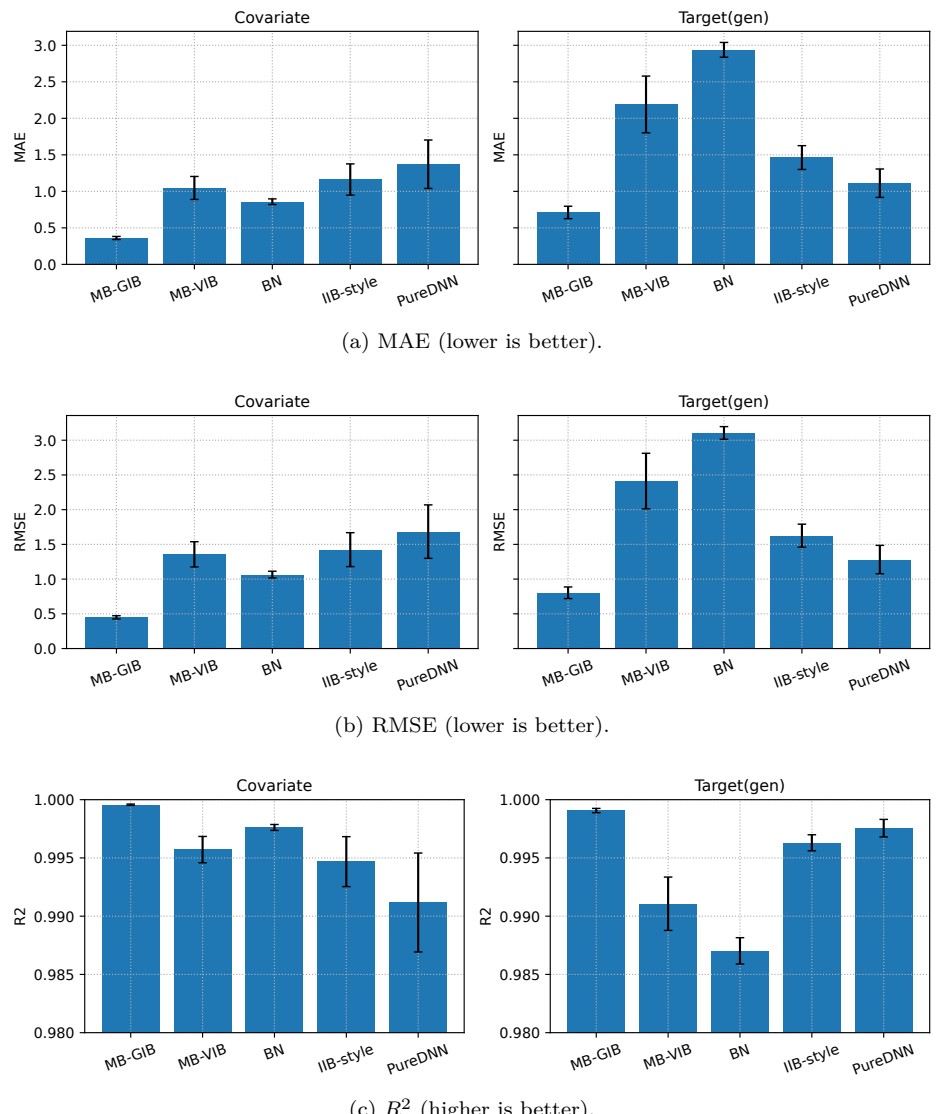

(a) MAE (lower is better).

(b) RMSE (lower is better).

(c) $R^2$ (higher is better).

Figure 3: **Main comparison under covariate and generalized target shift.** Bars show mean $\pm$ s.e. over seeds for MB–GIB, MB–VIB, BN, IIB-style, and pure DNN. In this well-specified linear–Gaussian SEM, MB–GIB leads on all metrics and the gap is largest under target shift; as discussed in the text, this reflects exact model match rather than a benefit of the IB objective per se.

**Capacity–compression tradeoffs in MB–VIB.** From the heatmaps (lower RMSE is better), the Student-$t$ setting favors a small latent capacity with stronger compression: the best performance occurs at $z_{\mathrm{dim}} = 4$ and $\beta = 0.01$. As capacity grows, $\beta$ must be tuned carefully—$z = 8, \beta = 0.003$ is competitive, but $z = 8, \beta = 0.01$ and $z = 16, \beta \in \{0.001, 0.01\}$ deteriorate markedly—consistent with heavy tails benefiting from aggressive compression to suppress outliers, while either too much compression (high $\beta$) or too little (low $\beta$) at large $z$ harms signal retention. Under Laplace noise, a broad low-RMSE band appears for moderate/large capacities $z \in [8, 16]$ with $\beta \in [0.003, 0.01]$; in contrast, $z = 4, \beta = 0.001$ is a clear failure mode. Overall, the results indicate a sweet-spot *coupling* between capacity and compression rather than a single universally optimal choice: heavier tails call for stronger compression and/or smaller $z$, whereas milder

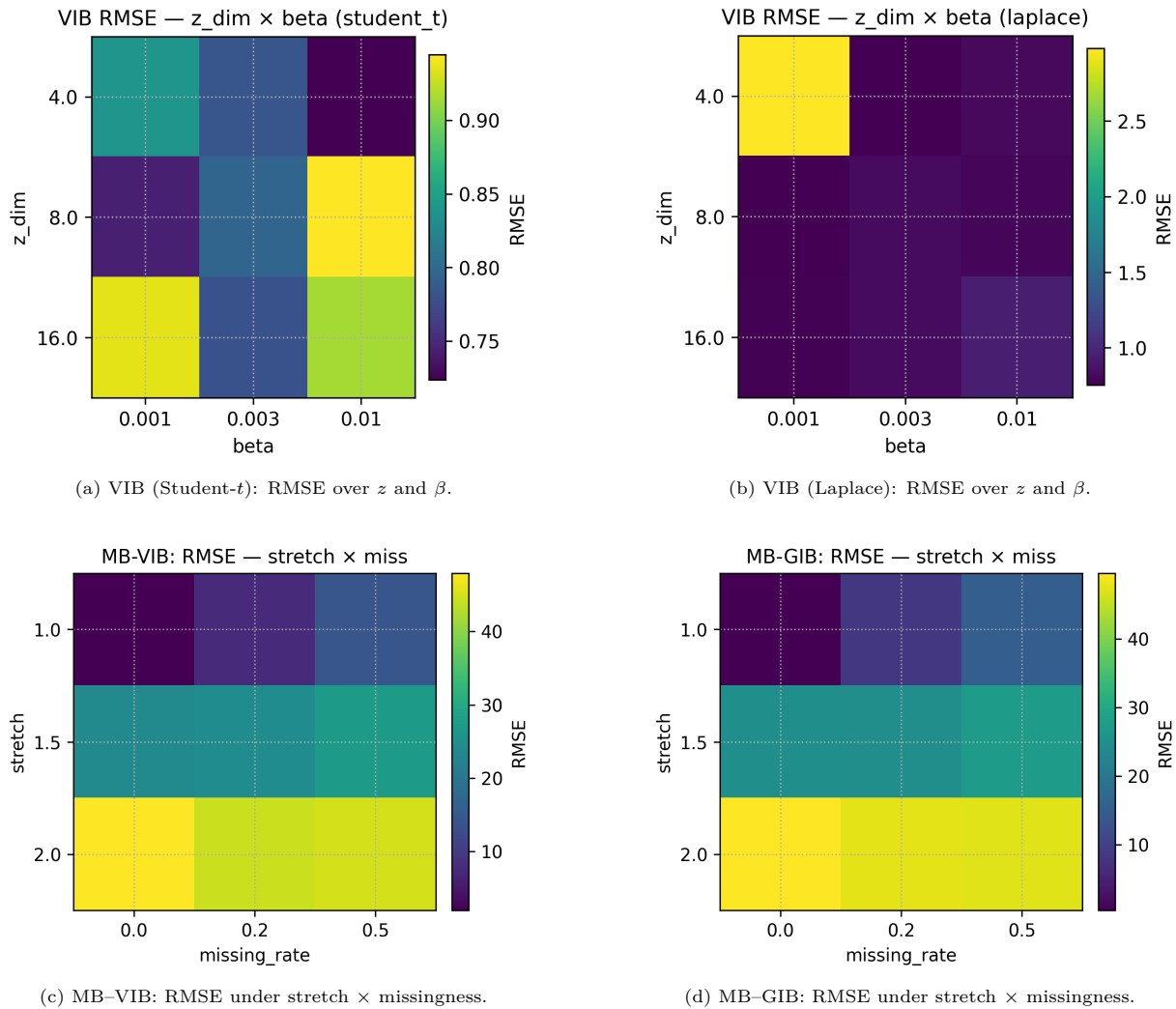

(a) VIB (Student-$t$): RMSE over $z$ and $\beta$.

(b) VIB (Laplace): RMSE over $z$ and $\beta$.

(c) MB–VIB: RMSE under stretch × missingness.

(d) MB–GIB: RMSE under stretch × missingness.

Figure 4: Heatmap summaries. **Top:** Ablation of VIB capacity ($z$) and compression ($\beta$) for two likelihoods. **Bottom:** Sensitivity to support mismatch (stretch) and missing rate for MB–VIB and MB–GIB. Lower is better (RMSE).

tails work best with moderate compression and larger $z$. Practically, one should start with a grid such as $z \in \{4, 8, 16\}$, $\beta \in \{0.003, 0.01\}$; if validation error rises when increasing $z$, raise $\beta$ slightly, and if error rises when increasing $\beta$, reduce $\beta$ or $z$. In short, MB–VIB performs best when capacity and compression are balanced to the noise regime—compress harder with heavy tails or small $z$, and use moderate $\beta$ with larger $z$ under milder tails.

**Sensitivity to support mismatch and missingness.** Figure 4 (c), (d) shows that prediction error is dominated by support mismatch rather than missingness. When the stretch factor $s$ (measuring covariate support shift) is near unity, both MB–VIB and MB–GIB achieve low RMSE, with a mild, roughly monotone degradation as the missingness rate $m$ rises from 0 to 0.5. Increasing the mismatch to $s = 1.5$ produces a uniform upward shift in RMSE across all $m$, indicating that imputation/regularization helps but cannot fully compensate for distributional shift. At severe mismatch $s = 2.0$, error becomes large and essentially insensitive to $m$, consistent with an extrapolation regime where the predictor leaves the training support. The two variants exhibit nearly identical profiles, suggesting that, for these settings, robustness is limited more by covariate support alignment than by the choice of bottleneck (MB–VIB vs. MB–GIB). Practically,

this argues for prioritizing support-alignment techniques (e.g., reweighting, representation alignment, or targeted data collection) before tackling high missingness rates; once support is aligned ($s \approx 1$), the methods remain usable even with substantial missingness ($m \leq 0.5$).

**Sensitivity to Markov-blanket misspecification.** Because the entire method hinges on access to the correct blanket, we report a blanket-misspecification study in the main text. On the same 7-node SEM under covariate shift (10 seeds), we perturb the encoder's input set in two ways: *drop-one*, which removes a true blanket variable (an omitted parent/child/spouse), and *add-non-MB*, which injects an extra non-blanket variable. Table 2 reports the result against the well-specified blanket. Two patterns emerge and both are consistent with the linear–Gaussian theory. Adding a non-blanket variable is essentially harmless ($R^2$ unchanged at 0.9996 for MB–GIB), exactly as the lossless-restriction result (Theorem 1) predicts: a conditionally independent extra input carries no additional information about $T$ and does not perturb the CCA spectrum. Dropping a true blanket variable, by contrast, degrades both estimators (MB–GIB RMSE rises from 0.449 to 0.755; MB–VIB from 1.40 to 2.07), because the omitted variable opens a path through which the shift can leak into the prediction. The practical implication, reflected in our recommendations, is asymmetric: when the blanket is uncertain it is safer to err toward an over-inclusive candidate set (and let the bottleneck compress) than to risk omitting a genuine blanket member.

Table 2: Sensitivity to Markov-blanket misspecification on the 7-node SEM (covariate shift, mean over 10 seeds; standard errors in parentheses). *Add-non-MB* injects a non-blanket variable; *drop-one* omits a true blanket variable. Dropping a true blanket member degrades performance, whereas adding a non-blanket variable is essentially harmless, as predicted by Theorem 1.

| Method | Blanket | MAE | RMSE | $R^2$ |
|--------|---------|-----|------|-------|
| MB-GIB | well-specified | 0.362 (0.021) | 0.449 (0.023) | 0.9996 |
|        | add-non-MB | 0.368 (0.021) | 0.457 (0.024) | 0.9996 |
|        | drop-one | 0.605 (0.018) | 0.755 (0.024) | 0.9988 |
| MB-VIB | well-specified | 1.154 (0.202) | 1.396 (0.235) | 0.9944 |
|        | add-non-MB | 1.308 (0.160) | 1.601 (0.188) | 0.9939 |
|        | drop-one | 1.632 (0.233) | 2.071 (0.277) | 0.9895 |

## 5.2 Simulated Experiments: MAGIC–IRRI Gene Network

We next consider a large-scale stress test on the 64-node *MAGIC–IRRI* Gaussian Bayesian network. To emulate strong experimental perturbations in a multi-trait genetic model, we apply three marginal-shift interventions on continuous covariates: **G4156** is shifted from $\mathcal{N}(0.7636, 0.9721^2)$ to $\mathcal{N}(1.5, 2.0^2)$, **G4573** from $\mathcal{N}(0.1196, 0.4744^2)$ to $\mathcal{N}(1.0, 1.0^2)$, and **G1533** from $\mathcal{N}(0.8004, 0.9803^2)$ to $\mathcal{N}(0, 3.0^2)$. After inducing these large shifts in the target domain, we hide the trait of interest HT and evaluate imputation from the remaining variables using several approaches: a source-trained Bayesian network (BN), mean/variance bottleneck variants (MB–GIB and MB–VIB), an IIB-style objective, and a pure feedforward DNN. All models are trained on the unshifted source and deployed zero-shot to the shifted domain.

**Results and analysis.** Table 3 sharpens the picture of where the blanket restriction helps and where the IB objective does. The headline observation is the large gap between every blanket-restricted method (top block plus MB–GIB/MB–VIB/IIB-style) and the two unrestricted baselines: the source-trained Bayesian network degrades into the out-of-distribution regime ($R^2 = -0.096$) and the unrestricted pure DNN fails outright ($R^2 = -1.77$), whereas a plain linear regression on MB($T$) already attains $R^2 = 0.569$. The blanket restriction itself—not the estimator placed on top of it—is therefore the primary driver of transfer robustness here. Within the blanket-restricted methods, the IB objective brings no accuracy advantage in this well-specified linear–Gaussian setting: MB–GIB (MAE 5.57, RMSE 7.01, $R^2 = 0.567$) is statistically indistinguishable from tuned linear/ridge regression on MB($T$) (MAE 5.58, RMSE 7.02, $R^2 = 0.569$), exactly

as the lossless-restriction result predicts since MB–GIB reduces to CCA in this regime. The variational MB–VIB (MAE 7.08, RMSE 10.02, $R^2 = 0.121$) and the IIB-style variant perform similarly to each other but trail both the linear blanket regressors and the tuned blanket MLP (MAE 6.80, $R^2 = 0.318$): the variational bottleneck can over-compress when the model is well-specified and a closed form is available. We read this as direct evidence for our revised framing—blanket restriction is the core transfer mechanism, while the IB objective is a principled estimator/regularizer whose benefit lies in nonlinear and non-Gaussian regimes where no closed-form conditional estimator exists (cf. Section 5 below). Overall, in high-dimensional but near-Gaussian causal networks, a cheap blanket regression or MB–GIB is the recommended default.

Table 3: Imputation performance on the MAGIC-IRRI DAG under multiple large-shift interventions. The top block reports tuned direct baselines that predict $T$ from MB($T$) *without* an IB objective (hyperparameters selected by source-side validation); the bottom block reports our methods and the unrestricted baselines. In this well-specified linear–Gaussian regime, OLS-type blanket regressors and MB–GIB are statistically indistinguishable, while all blanket-restricted methods dominate the unrestricted Bayesian network and DNN.

| Method | MAE | RMSE | $R^2$ |
|---|---|---|---|
| *Direct blanket regressors (no IB objective)* | | | |
| LinearReg (MB only) | 5.580 | 7.019 | 0.569 |
| Ridge (MB only, tuned) | 5.580 | 7.019 | 0.569 |
| KernelRidge (MB only, tuned) | 5.659 | 7.112 | 0.558 |
| MLP (MB only, tuned) | 6.795 | 8.825 | 0.318 |
| *Bottleneck methods and unrestricted baselines* | | | |
| MB-GIB | 5.5706 | 7.0083 | 0.5670 |
| MB-VIB | 7.0837 | 10.0190 | 0.1211 |
| IIB-style | 7.8219 | 10.0456 | 0.1165 |
| Bayesian Network (unrestricted) | 9.3827 | 11.1872 | $-0.0957$ |
| Pure DNN (unrestricted) | 14.4523 | 17.7908 | $-1.7711$ |

### 5.3 Real-Data Experiment on Single-Cell Signaling Networks

To assess performance in a setting with genuine biological heterogeneity, we use the seminal single-cell flow–cytometry compendium of Sachs *et al.* (Sachs et al., 2005), which quantifies phosphorylation levels for a panel of signaling proteins in human primary CD4$^+$ T cells subjected to controlled perturbations. The experimental conditions in this dataset create natural distributional shifts, providing a robust testbed for causal transfer. In our study, the anti–CD3/CD28 stimulation (853 cells) serves as the *source* domain, while the phorbol ester PMA condition (913 cells) is treated as the *target*. The latter directly activates PKC and reshapes downstream pathways in ways not present under CD3/CD28, yielding a demanding transfer scenario. With this split fixed, we evaluate models on imputing/forecasting the activities of ten key nodes in the pathway: Raf, Mek, Plcg, PIP$_2$, PIP$_3$, Erk, Akt, PKA, P38, and Jnk. Quantitative results—reported as mean absolute error (MAE), root mean squared error (RMSE), and coefficient of determination ($R^2$)—are summarized in Table 4.

Across the ten targets, bottleneck-based imputers are consistently strong under multiple interventions. In particular, MB-GIB attains the best or tied-best scores on *Raf*, *Erk*, and *PKA*, delivering large error reductions and high $R^2$ (e.g., $R^2$=0.9184 on *Erk*). MB-VIB is competitive—most notably it achieves the best RMSE and $R^2$ on *PIP2* and near-best on *PIP3*. IIB-style excels on *Plcg* where it achieves the top performance across all three metrics. By contrast, the Pure DNN lags on several targets and frequently yields negative $R^2$, indicating overfitting or poor robustness to distribution shift induced by interventions.

The Bayesian Network (BN) remains a strong baseline on some nodes, tying for best on *Mek*, *Akt*, *P38*, and *Jnk*. On these targets the BN and MB-GIB entries coincide to all reported digits, which is expected rather than coincidental: when the local conditional $p(T \mid \mathrm{MB}(T))$ is well-approximated by a linear–Gaussian law, the BN's fitted local conditional mean and MB-GIB's OLS-on-blanket decoder are the same estimator, so

Table 4: Imputation performance on the Sachs et al. data under multiple interventions.

| Target | Method | MAE | RMSE | R2 | Target | Method | MAE | RMSE | R2 |
|---|---|---|---|---|---|---|---|---|---|
| Raf | Bayesian Network | 0.6908 | 1.0015 | -0.0041 | Erk | Bayesian Network | 0.5884 | 0.8379 | 0.2971 |
| | MB-GIB | **0.4132** | **0.6393** | **0.5908** | | MB-GIB | **0.1817** | **0.2854** | **0.9184** |
| | MB-VIB | 0.4562 | 0.6821 | 0.5343 | | MB-VIB | 0.3527 | 0.402 | 0.7691 |
| | IIB-style | 0.4422 | 0.6840 | 0.5316 | | IIB-style | 0.424 | 0.5919 | 0.6496 |
| | Pure DNN | 0.5205 | 0.7815 | 0.3885 | | Pure DNN | 0.3397 | 0.4499 | 0.7973 |
| Mek | Bayesian Network | **0.3935** | **0.6385** | **0.5919** | Akt | Bayesian Network | **0.1744** | **0.2756** | **0.9239** |
| | MB-GIB | **0.3935** | **0.6385** | **0.5919** | | MB-GIB | **0.1744** | **0.2756** | **0.9239** |
| | MB-VIB | 0.4306 | 0.7032 | 0.5050 | | MB-VIB | 0.2186 | 0.3014 | 0.9090 |
| | IIB-style | 0.5348 | 1.0905 | -0.1904 | | IIB-style | 0.3901 | 0.4595 | 0.7885 |
| | Pure DNN | 0.7599 | 1.2648 | -0.6014 | | Pure DNN | 0.2475 | 0.3505 | 0.8769 |
| Plcg | Bayesian Network | 0.6529 | 0.9995 | -0.0000 | PKA | Bayesian Network | 0.6810 | 0.9992 | 0.0006 |
| | MB-GIB | 0.5858 | 0.9008 | 0.1876 | | MB-GIB | **0.5114** | **0.7376** | **0.4552** |
| | MB-VIB | 0.5326 | 0.8075 | 0.3472 | | MB-VIB | 1.6468 | 2.3798 | -4.6698 |
| | IIB-style | **0.444** | **0.6999** | **0.5095** | | IIB-style | 0.8304 | 1.5230 | -1.3221 |
| | Pure DNN | 0.6443 | 1.0064 | -0.0139 | | Pure DNN | 1.0835 | 1.5752 | -1.4842 |
| PIP2 | Bayesian Network | 0.6156 | 0.8253 | 0.3179 | P38 | Bayesian Network | 0.2896 | **0.4542** | **0.7934** |
| | MB-GIB | 0.6156 | 0.8254 | 0.3179 | | MB-GIB | 0.2896 | **0.4542** | **0.7934** |
| | MB-VIB | 0.4976 | **0.7609** | **0.4203** | | MB-VIB | **0.2801** | **0.4542** | **0.7934** |
| | IIB-style | 0.5158 | 0.9556 | 0.0856 | | IIB-style | 0.3087 | 0.448 | 0.7647 |
| | Pure DNN | **0.4655** | 0.7948 | 0.3675 | | Pure DNN | 0.2850 | 0.4609 | 0.7873 |
| PIP3 | Bayesian Network | 0.5240 | 0.9280 | 0.1378 | Jnk | Bayesian Network | **0.6621** | **1.1185** | **-0.2525** |
| | MB-GIB | 0.3809 | **0.8106** | **0.3421** | | MB-GIB | **0.6621** | **1.1185** | **-0.2525** |
| | MB-VIB | **0.3489** | 0.8307 | 0.3090 | | MB-VIB | 0.6784 | 1.1327 | -0.2844 |
| | IIB-style | 0.5336 | 1.2618 | -0.5939 | | IIB-style | 0.6784 | 1.1347 | -0.2889 |
| | Pure DNN | 0.5219 | 1.3457 | -0.8131 | | Pure DNN | 0.7111 | 1.1530 | -0.3310 |

they return identical imputations. More generally, this indicates that when local dependencies are simple or close to Gaussian/linear, a structured generative model is already sufficient. However, on targets where intervention-induced heterogeneity is more pronounced (*Raf*, *Erk*, *PKA*), mean-aware bottlenecking (MB-GIB) offers clear gains, pointing to better invariance and regularization under shift. Discrepancies between MAE and RMSE (e.g., *PIP2*) imply differing outlier sensitivity across methods; MB-VIB's RMSE lead there suggests improved handling of rare but large errors. Overall, the results favor MB-GIB as the most reliable across interventions, with MB-VIB and IIB-style providing complementary strengths on specific targets, and BN serving as a competitive fallback where structure is well aligned with the data.

### 5.4 Extensions and Limitations

*Extensions.* The framework naturally accommodates (i) *partial or learned blankets*: estimate a candidate $MB(T)$ from local discovery around $T$ and prune by conditional-independence tests; (ii) *parents-only* encoders when downstream paths are unstable; (iii) *multi-environment* training by sharing the decoder $q_\theta(t \mid z)$ across environments while allowing environment-specific encoders or priors; (iv) *semi-supervised* target settings with a small labeled subset to fine-tune the decoder; and (v) *uncertainty quantification* via the VIB decoder's predictive variance/entropy and MB–GIB's linear-Gaussian posterior formulas.

*Limitations.* Performance hinges on the *MB-invariance* assumption; mechanism drift in $p(T \mid X_M)$ (e.g., new parents of $T$, altered noise law) breaks zero-shot guarantees. Mis-specified or incomplete blankets (omitted true parents/spouses) can leak shift through $X \setminus M$ or reduce efficiency. Severe support shift in $X_M$ induces extrapolation regardless of model class. VIB requires tuning $(d_z, \beta)$ and sufficient data; over-capacity can overfit nuisance within $M$, while under-capacity can underfit $p(T \mid X_M)$. We recommend routine diagnostics: compare source vs. target decoder log-likelihoods and residuals of $\hat{g}(X_{M,t})$ to flag violations; if detected, shrink to parents-only, increase $\beta$ (more compression), or incorporate limited target labels to recalibrate the decoder. *Finite-sample/optimization:* MB–GIB requires well-conditioned covariances; MB–VIB needs capacity control and early stopping. *Latent confounding:* Hidden parents of $T$ that also affect $X_M$ can violate

conditional independence assumptions; instrumental/negative-control extensions are a promising avenue but beyond our scope.

## 6 Conclusion

We presented a *DAG-aware Information Bottleneck* for causal domain adaptation when the target variable is entirely missing at deployment, in the setting where local causal structure around the target is known or estimable and the target mechanism is stable across domains. The transfer mechanism is structural: restricting the predictor to the Markov blanket of the target screens off shift-prone non-blanket variation and yields zero-shot transfer under blanket invariance. On top of this restriction we use the Information Bottleneck as a principled estimator—MB–GIB, a closed-form CCA-equivalent solution with a lossless-restriction guarantee in the linear-Gaussian regime, and MB–VIB for nonlinear or non-Gaussian data. Our experiments make the division of labor explicit: across a controlled 7-node SEM, the 64-node MAGIC–IRRI network, and a single-cell signaling dataset, blanket-restricted methods are uniformly robust under large covariate and generalized target shifts, far outperforming unrestricted baselines; within blanket-restricted methods, the IB objective brings no accuracy gain over plain blanket regression in well-specified linear-Gaussian regimes (where it can even over-compress), and its benefit as a flexible, regularized estimator emerges in nonlinear and non-Gaussian regimes where no closed-form conditional estimator is available.

Beyond performance, the practical recipe is simple: (i) restrict encoders to parents or the Markov blanket, (ii) choose MB–GIB for fast, closed-form estimation or MB–VIB for flexible likelihoods, and (iii) deploy zero-shot to the target. Routine diagnostics (residual stability, likelihood checks) help detect violations of MB-invariance; when detected, shrinking to parents-only, increasing compression, or leveraging a small set of target labels restores reliability.

This work turns local causal knowledge into a lightweight, scalable toolkit for imputation under shift. Future directions include handling mechanism drift inside the blanket, learning blankets with uncertainty, incorporating instrumental/negative controls for latent confounding, and extending to multi-environment training with shared decoders and environment-specific encoders. We hope these results encourage broader use of mechanism-stable, bottlenecked representations as a practical bridge between causal structure and real-world domain adaptation.

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

# A    Appendix: Theoretical Results and Proofs For main Claims in Sections 4.1 and 4.2

**MB–GIB is Globally Optimal (Linear–Gaussian / CCA)**

We restate the lossless-restriction result of Section 4.1 as a standalone statement and then prove it. The argument uses only Gaussian conditional independence and the CCA/GIB equivalence.

**Theorem 5** (Lossless blanket restriction; full statement). *Let $(X, T)$ be jointly Gaussian with $X = (M, N)$, where $M = \mathrm{MB}(T)$ and $N = X \setminus M$, and assume $T \perp\!\!\!\perp N \mid M$. Let*

$$\Omega_X := \Sigma_{XX}^{-1/2} \Sigma_{XT} \Sigma_{TT}^{-1} \Sigma_{TX} \Sigma_{XX}^{-1/2}, \qquad \Omega_M := \Sigma_{MM}^{-1/2} \Sigma_{MT} \Sigma_{TT}^{-1} \Sigma_{TM} \Sigma_{MM}^{-1/2}$$

*be the global and blanket CCA/IB operators. Then (i) the nonzero spectra of $\Omega_X$ and $\Omega_M$ coincide; (ii) every eigenvector of $\Omega_M$ with nonzero eigenvalue lifts to an eigenvector of $\Omega_X$ with the same eigenvalue via a column-orthonormal map $Q$; (iii) for every bottleneck dimension $d$, the optimal $d$-dimensional IB/CCA subspaces of $(X, T)$ and $(M, T)$ are isometric, so restricting the encoder to $M$ incurs no loss; and (iv) $\mathbb{E}[T \mid X] = \mathbb{E}[T \mid M]$, hence the Bayes risk of predicting $T$ from $X$ equals that from $M$.*

*Proof.* The proof proceeds through the block-matrix factorization below.

*Block notation and the MB identity.* Partition $X = (M, N)$ and write

$$\Sigma_{XX} = \begin{bmatrix} \Sigma_{MM} & \Sigma_{MN} \\ \Sigma_{NM} & \Sigma_{NN} \end{bmatrix}, \qquad \Sigma_{XT} = \begin{bmatrix} \Sigma_{MT} \\ \Sigma_{NT} \end{bmatrix}.$$

For Gaussians, $T \perp\!\!\!\perp N \mid M$ is equivalent to the vanishing conditional cross-covariance

$$\Sigma_{NT \cdot M} := \Sigma_{NT} - \Sigma_{NM} \Sigma_{MM}^{-1} \Sigma_{MT} = 0,$$

hence

$$\Sigma_{NT} = \Sigma_{NM} \Sigma_{MM}^{-1} \Sigma_{MT}. \tag{1}$$

*Factorization of $\Sigma_{XT}$ via $M$.* Let

$$L := \begin{bmatrix} I_{|M|} \\ B \end{bmatrix}, \qquad B := \Sigma_{NM} \Sigma_{MM}^{-1}.$$

Then equation 1 yields

$$\Sigma_{XT} = \begin{bmatrix} \Sigma_{MT} \\ \Sigma_{NT} \end{bmatrix} = \begin{bmatrix} I \\ B \end{bmatrix} \Sigma_{MT} = L \, \Sigma_{MT}. \tag{2}$$

*CCA operators and a key metric identity.* Define the (symmetric) CCA/IB operators

$$\Omega_X := \Sigma_{XX}^{-1/2} \Sigma_{XT} \Sigma_{TT}^{-1} \Sigma_{TX} \Sigma_{XX}^{-1/2}, \qquad \Omega_M := \Sigma_{MM}^{-1/2} \Sigma_{MT} \Sigma_{TT}^{-1} \Sigma_{TM} \Sigma_{MM}^{-1/2}.$$

Using equation 2,

$$\Omega_X = \left( \Sigma_{XX}^{-1/2} L \right) \underbrace{\left( \Sigma_{MT} \Sigma_{TT}^{-1} \Sigma_{TM} \right)}_{=:K} \left( \Sigma_{XX}^{-1/2} L \right)^\top.$$

A standard block-inverse (Schur-complement) calculation under $\Sigma_{NT \cdot M} = 0$ gives the metric identity

$$L^\top \Sigma_{XX}^{-1} L = \Sigma_{MM}^{-1}. \tag{3}$$

Equivalently,

$$\Sigma_{XX}^{-1/2} L = Q \Sigma_{MM}^{-1/2} \quad \text{for some column-orthonormal } Q \; (Q^\top Q = I_{|M|}).$$

*Same canonical correlations via SVD.* Let

$$S_X := \Sigma_{XX}^{-1/2}\Sigma_{XT}\Sigma_{TT}^{-1/2} = \left(\Sigma_{XX}^{-1/2}L\right)\left(\Sigma_{MT}\Sigma_{TT}^{-1/2}\right) = Q\underbrace{\left(\Sigma_{MM}^{-1/2}\Sigma_{MT}\Sigma_{TT}^{-1/2}\right)}_{=:S_M}.$$

Since $Q$ has orthonormal columns, left-multiplication by $Q$ preserves singular values, hence

$$\mathrm{singvals}(S_X) = \mathrm{singvals}(S_M).$$

Therefore the canonical correlations for $(X,T)$ equal those for $(M,T)$, and the nonzero spectra of $\Omega_X = S_X S_X^\top$ and $\Omega_M = S_M S_M^\top$ coincide.

*Range containment and lifted eigenvectors.* From the factorization,

$$\mathrm{range}(\Omega_X) \subseteq \mathrm{range}\left(\Sigma_{XX}^{-1/2}L\right) = \mathrm{range}\left(Q\,\Sigma_{MM}^{-1/2}\right) = \left\{\,\Sigma_{XX}^{-1/2}\begin{bmatrix}u\\Bu\end{bmatrix} : u \in \mathbb{R}^{|M|}\right\}.$$

If $v_M$ is an eigenvector of $\Omega_M$ with a nonzero eigenvalue, then $v_X := Q\,v_M$ is an eigenvector of $\Omega_X$ with the same eigenvalue.

*Equivalence of optimal $d$-dimensional subspaces.* Let $\mathcal{S}_d(X)$ and $\mathcal{S}_d(M)$ be any optimal $d$-dimensional IB/CCA subspaces. Then

$$\mathcal{S}_d(X) = \Sigma_{XX}^{-1/2}L\,\Sigma_{MM}^{1/2}\,\mathcal{S}_d(M) = Q\,\mathcal{S}_d(M),$$

i.e., the optimal $X$-subspace is exactly the *isometric lift* of the optimal $M$-subspace. Hence restricting IB/CCA to the Markov blanket $M$ incurs no loss for any $d$.

*Prediction risk.* Since $\Sigma_{XT} = L\Sigma_{MT}$ and $L = [I; B]$, we have $\mathbb{E}[T \mid X] = \mathbb{E}[T \mid M]$ (the Schur-complement condition $\Sigma_{NT \cdot M} = 0$). Thus the Bayes MSE using $X$ equals that using $M$; the IB-optimal linear summaries agree up to the isometry $Q$. This establishes claims (i)–(iv). $\qquad\square$

### Justifying MB–VIB Beyond Gaussianity

We justify *MB–VIB* (using only the Markov blanket of $T$) well beyond Gaussianity with information-theoretic and causal arguments that do not depend on linearity or normality, and collect the population-level statement as Proposition 1.

### 1) Information-theoretic sufficiency (distribution-free)

Let $X = (M, N)$ where $M := \mathrm{MB}(T)$ and $N := X \setminus M$. In any DAG-Markov distribution (not necessarily Gaussian),

$$T \perp\!\!\!\perp N \mid M.$$

Then:

- **All predictive information is in $M$:**

  $$I(T;X) = I\left(T;(M,N)\right) = I(T;M) + I(T;N \mid M) = I(T;M),$$

  since $I(T;N \mid M) = 0$ by conditional independence.

- **IB objective cannot benefit from $N$:** The (variational) IB Lagrangian for any representation $U = f(X)$ is

  $$\mathcal{L}_\beta(f) = I(X;U) - \beta\,I(T;U).$$

  By the data processing inequality (DPI),

  $$I(T;U) \leq I(T;X) = I(T;M).$$

  Moreover, any $U$ that is a function of $M$ can achieve the same upper bound on $I(T;U)$ as any $U$ that also sees $N$ (since $N$ contains no extra information about $T$ given $M$). But using $N$ can only *increase* the compression cost $I(X;U)$ (the encoder ingests more input), so among encoders with the same $I(T;U)$, those depending only on $M$ weakly *dominate* in $\mathcal{L}_\beta$.

**Proposition 1** (Distribution-free blanket sufficiency for the IB objective)**.** *Let $X = (M, N)$ with $M = \mathrm{MB}(T)$ and $N = X \setminus M$, and suppose $T \perp\!\!\!\perp N \mid M$ in any (not necessarily Gaussian) DAG-Markov distribution. Then $I(T; X) = I(T; M)$ and $p(T \mid X) = p(T \mid M)$. Consequently, if the encoder class is rich enough to approximate measurable functions, then for every $\beta > 0$ there exists an IB-optimal encoder $f^\star$ for the Lagrangian $\mathcal{L}_\beta(f) = I(X; U) - \beta I(T; U)$ that depends only on $M$; i.e., restricting the IB encoder to the Markov blanket does not worsen the optimal IB value.*

*Proof.* For any encoder $f(M, N)$, define $g(M) = \mathbb{E}[f(M, N) \mid M]$ (or a sufficient surrogate achieving the same $I(T; U)$). By the data processing inequality and $T \perp\!\!\!\perp N \mid M$, the value $I(T; g(M)) \geq I(T; f(M, N))$ is achievable, while $I(X; g(M)) \leq I(X; f(M, N))$. Hence $\mathcal{L}_\beta(g) \leq \mathcal{L}_\beta(f)$, so an $M$-measurable encoder weakly dominates. □

## 2) Predictive optimality via conditional sufficiency

Independently of IB, the Bayes predictor satisfies

$$p(T \mid X) \;=\; p(T \mid M),$$

so the *minimal sufficient statistic* for predicting $T$ is any representation that preserves $p(T \mid M)$. Thus, if the VIB encoder/decoder can approximate $p(T \mid U)$, then any VIB-optimal representation can be chosen to be a function of $M$; including $N$ brings no improvement to (population) predictive risk.

## 3) Robustness to shift and interventions (causal justification)

- Under *covariate shift* where $p(X)$ changes but the mechanism $p(T \mid \mathrm{pa}(T))$ stays fixed, non-blanket variables $N$ can drift via $p(N \mid M)$ without changing $p(T \mid M)$. Using $N$ risks encoding unstable directions; restricting to $M$ preserves the invariant conditional.

- Under *soft interventions* on variables outside $\mathrm{MB}(T)$, $p(T \mid M)$ remains invariant by the causal Markov property, while $p(T \mid X)$ (if it exploits $N$) can change. MB–VIB therefore targets invariant predictive structure.

## 4) Beyond linear/Gaussian encoders (practical learnability)

- With nonlinear VIB (neural encoders/decoders), universal approximation allows a representation $U = \phi(M)$ that is (approximately) sufficient for $T$ without using $N$. This holds for non-Gaussian and nonlinear SEMs so long as $T \perp\!\!\!\perp N \mid M$.

- With finite samples and model/regularization constraints, MB–VIB often helps by reducing variance and avoiding spurious correlations—another reason it can outperform global inputs under shift.

## 5) When MB–VIB might not be optimal

- If the Markov blanket is *misspecified* (e.g., hidden confounders or measurement error breaking $T \perp\!\!\!\perp N \mid M$), some components of $N$ may carry residual predictive information; global representations can help.

- If only a *subset* of $M$ is observed (missing parents/children/spouses), certain $N$ variables may act as proxies; the MB restriction could be too strict.

- With a *tight* encoder class (very low capacity or heavy regularization), allowing $N$ might accidentally aid approximation—even though it adds no information in principle.

**Bottom line (distribution-free)**

1. **Sufficiency:** $T \perp\!\!\!\perp N \mid M \Rightarrow I(T;X) = I(T;M)$ and $p(T \mid X) = p(T \mid M)$.

2. **IB dominance:** For the IB objective $I(X;U) - \beta I(T;U)$, any benefit to $I(T;U)$ achievable with $X$ is achievable with $M$ alone, while $I(X;U)$ cannot be smaller when using superfluous inputs.

3. **Invariance:** MB focuses the encoder on causally relevant mechanisms, the ones most likely to remain stable across domains.

None of these require Gaussianity or linearity; they rely only on conditional independence and data processing / information sufficiency.

In this context, **whitening** means linearly transforming a zero-mean vector so its covariance becomes the identity. If $X \in \mathbb{R}^q$ has mean $\mu_X$ and covariance $\Sigma_{XX} \succ 0$, a (symmetric) whitener is $\Sigma_{XX}^{-1/2}$ (from an eigendecomposition or Cholesky). The whitened variable is

$$\tilde{X} = \Sigma_{XX}^{-1/2}, (X - \mu_X), \qquad \mathrm{Cov}(\tilde{X}) = I_q.$$

Intuitively, whitening de-correlates the coordinates and rescales each to unit variance.

# B  Appendix: Theoretical Results and Proofs For main Claims in Section 4.4

*Proof of Theorem 2.* Let $d := p + 1$ and let $Z_1, \ldots, Z_n \in \mathbb{R}^d$ be i.i.d. copies of $Z = (X_M^\top, T)^\top$ with $\mathbb{E}(Z) = 0$ and covariance $\Sigma = \mathrm{Cov}(Z) = \mathbb{E}[ZZ^\top]$. For simplicity, define the uncentered second-moment estimator

$$\widehat{\Sigma} := \frac{1}{n} \sum_{i=1}^n Z_i Z_i^\top.$$

The centered sample covariance $\widehat{\Sigma}_c := \frac{1}{n} \sum_{i=1}^n (Z_i - \bar{Z})(Z_i - \bar{Z})^\top$ satisfies $\widehat{\Sigma}_c = \widehat{\Sigma} - \bar{Z}\bar{Z}^\top$. Moreover, $\|\bar{Z}\bar{Z}^\top\|_{\mathrm{op}} = \|\bar{Z}\|_2^2$ and a standard sub-Gaussian mean bound yields $\|\bar{Z}\|_2 \le CK\left(\sqrt{\frac{d+\log(1/\delta)}{n}} + \frac{d+\log(1/\delta)}{n}\right)$ with probability at least $1 - \delta$, hence $\|\bar{Z}\bar{Z}^\top\|_{\mathrm{op}} \le C'K^2 \frac{d+\log(1/\delta)}{n}$. This term is dominated by the stated rate, so the same bound holds for $\widehat{\Sigma}_c$ up to constants.

Under Assumption 1, the random vector $Z$ is $K$-sub-Gaussian. A standard non-asymptotic covariance estimation theorem for general (non-isotropic) sub-Gaussian vectors (e.g., Vershynin, *High-Dimensional Probability*, 2018, covariance estimation for sub-Gaussian vectors; see Theorem 4.7.1 in the Cambridge edition) implies that for any $\delta \in (0, 1)$, with probability at least $1 - \delta$,

$$\|\widehat{\Sigma} - \Sigma\|_{\mathrm{op}} \le cK^2\left(\sqrt{\frac{d + \log(1/\delta)}{n}} + \frac{d + \log(1/\delta)}{n}\right),$$

for a universal constant $c > 0$. Substituting $d = p + 1$ yields the stated bound.

Finally, the same high-probability bound holds for each covariance block. For any principal block $B$ of a symmetric matrix $A$, $\|B\|_{\mathrm{op}} \le \|A\|_{\mathrm{op}}$. For the cross-covariance block, partitioning indices by $(X_M, T)$, we have

$$\|A_{X_M T}\|_{\mathrm{op}} = \sup_{\|u\|_2 = \|v\|_2 = 1} u^\top A_{X_M T} v = \sup_{\|u\|_2 = \|v\|_2 = 1} \tilde{u}^\top A \tilde{v} \le \|A\|_{\mathrm{op}},$$

where $\tilde{u} = (u^\top, 0)^\top$ and $\tilde{v} = (0, v)^\top$ are unit vectors in $\mathbb{R}^d$ and $u \in \mathbb{R}^p$ and $v \in \mathbb{R}$ since $T$ is scalar.

Applying this to $A = \widehat{\Sigma} - \Sigma$ completes the proof. $\square$

*Proof of Theorem 3.* Recall

$$\Omega_M = \Sigma_{XX}^{-1/2} \Sigma_{XT} \Sigma_{TT}^{-1} \Sigma_{TX} \Sigma_{XX}^{-1/2}, \qquad \widehat{\Omega}_M = \widehat{\Sigma}_{XX}^{-1/2} \widehat{\Sigma}_{XT} \widehat{\Sigma}_{TT}^{-1} \widehat{\Sigma}_{TX} \widehat{\Sigma}_{XX}^{-1/2}.$$

Throughout this proof, we abbreviate $\Sigma_{X_M X_M}$ as $\Sigma_{XX}$ for notational simplicity, with the understanding that this refers to the Markov blanket covariance. Also, we use the partition $Z = (X_M^\top, T)^\top$, so $\Sigma_{TT} \in \mathbb{R}$ is a scalar variance. Since $T$ is one-dimensional, we may (and do) assume w.l.o.g. that $\Sigma_{TT} = 1$ by rescaling $T$; this only rescales $K$ and does not affect the form of the rate.

*Davis–Kahan for symmetric eigen-subspaces.* Both $\Omega_M$ and $\widehat{\Omega}_M$ are symmetric positive semidefinite. Let $\lambda_1 \geq \cdots \geq \lambda_p$ be the eigenvalues of $\Omega_M$ and assume the eigengap $\gamma = \lambda_d - \lambda_{d+1} > 0$ (Assumption 3). The Davis–Kahan $\sin\Theta$ theorem (in operator norm form) yields

$$\|\sin\Theta(\widehat{U}, U_\star)\|_{\mathrm{op}} \leq \frac{C}{\gamma} \|\widehat{\Omega}_M - \Omega_M\|_{\mathrm{op}}, \tag{4}$$

for a universal constant $C > 0$. Moreover, the projector bound $\|\widehat{\Pi} - \Pi_\star\|_{\mathrm{op}} \leq 2\|\sin\Theta(\widehat{U}, U_\star)\|_{\mathrm{op}}$ is standard.

*Reduce $\|\widehat{\Omega}_M - \Omega_M\|_{\mathrm{op}}$ to covariance errors.* Write

$$S := \Sigma_{XX}^{-1/2}, \quad \widehat{S} := \widehat{\Sigma}_{XX}^{-1/2}, \qquad M := \Sigma_{XT}\Sigma_{TT}^{-1}\Sigma_{TX} = \Sigma_{XT}\Sigma_{TX}, \quad \widehat{M} := \widehat{\Sigma}_{XT}\widehat{\Sigma}_{TT}^{-1}\widehat{\Sigma}_{TX}.$$

Then $\Omega_M = SMS$ and $\widehat{\Omega}_M = \widehat{S}\widehat{M}\widehat{S}$, so

$$\widehat{\Omega}_M - \Omega_M = (\widehat{S} - S)\widehat{M}\widehat{S} + S(\widehat{M} - M)\widehat{S} + SM(\widehat{S} - S). \tag{5}$$

We work on the event

$$\mathcal{E} := \left\{ \|\widehat{\Sigma} - \Sigma\|_{\mathrm{op}} \leq \tfrac{\lambda_0}{2} \right\} \cap \left\{ |\widehat{\Sigma}_{TT} - \Sigma_{TT}| \leq \tfrac{1}{2}\Sigma_{TT} \right\}, \tag{6}$$

which holds with probability at least $1 - \delta$ for suitable constants by Theorem 2 (applied with a union bound to the relevant blocks). On $\mathcal{E}$,

$$\lambda_{\min}(\widehat{\Sigma}_{XX}) \geq \lambda_0/2, \qquad \lambda_{\max}(\widehat{\Sigma}_{XX}) \leq \Lambda_0 + \lambda_0/2, \qquad \widehat{\Sigma}_{TT} \in [1/2, 3/2].$$

Hence

$$\|S\|_{\mathrm{op}} \leq \lambda_0^{-1/2}, \qquad \|\widehat{S}\|_{\mathrm{op}} \leq (\lambda_0/2)^{-1/2} = \sqrt{2}\,\lambda_0^{-1/2}. \tag{7}$$

*Control $M, \widehat{M}$ and their perturbation.* Since the full covariance matrix of $(X_M, T)$ is positive semidefinite, its Schur complement implies

$$\Sigma_{XX} - \Sigma_{XT}\Sigma_{TT}^{-1}\Sigma_{TX} = \Sigma_{XX} - M \succeq 0,$$

so $0 \preceq M \preceq \Sigma_{XX}$ and therefore

$$\|M\|_{\mathrm{op}} \leq \|\Sigma_{XX}\|_{\mathrm{op}} \leq \Lambda_0. \tag{8}$$

Similarly, on $\mathcal{E}$, the empirical covariance is also positive semidefinite, giving $0 \preceq \widehat{M} \preceq \widehat{\Sigma}_{XX}$ (up to the scalar factor $\widehat{\Sigma}_{TT}^{-1} \in [2/3, 2]$), and thus

$$\|\widehat{M}\|_{\mathrm{op}} \leq 2\|\widehat{\Sigma}_{XX}\|_{\mathrm{op}} \leq 2(\Lambda_0 + \lambda_0/2) \leq 3\Lambda_0 \quad \text{(w.l.o.g. taking } \Lambda_0 \geq \lambda_0\text{).} \tag{9}$$

Next, bound $\|\widehat{M} - M\|_{\mathrm{op}}$. Using $\Sigma_{TT} = 1$ and adding/subtracting terms,

$$\widehat{M} - M = (\widehat{\Sigma}_{XT} - \Sigma_{XT})\,\Sigma_{TX} + \widehat{\Sigma}_{XT}\,(\widehat{\Sigma}_{TT}^{-1} - 1)\,\Sigma_{TX} + \widehat{\Sigma}_{XT}\widehat{\Sigma}_{TT}^{-1}(\widehat{\Sigma}_{TX} - \Sigma_{TX}).$$

Each covariance block deviation is bounded by $\|\widehat{\Sigma} - \Sigma\|_{\mathrm{op}}$ (as shown in the block argument in Theorem 2). Also, from $M \preceq \Sigma_{XX}$ we have $\|\Sigma_{XT}\|_{\mathrm{op}}^2 = \|\Sigma_{XT}\Sigma_{TX}\|_{\mathrm{op}} = \|M\|_{\mathrm{op}} \leq \Lambda_0$, so $\|\Sigma_{XT}\|_{\mathrm{op}} \leq \sqrt{\Lambda_0}$; the same reasoning on $\mathcal{E}$ gives $\|\widehat{\Sigma}_{XT}\|_{\mathrm{op}} \leq \sqrt{3\Lambda_0}$. Finally, on $\mathcal{E}$, $|\widehat{\Sigma}_{TT}^{-1} - 1| \leq 2|\widehat{\Sigma}_{TT} - 1| \leq 2\|\widehat{\Sigma} - \Sigma\|_{\mathrm{op}}$. Putting these together yields, on $\mathcal{E}$,

$$\|\widehat{M} - M\|_{\mathrm{op}} \leq C_M \|\widehat{\Sigma} - \Sigma\|_{\mathrm{op}}, \tag{10}$$

for a constant $C_M > 0$ depending only on $\Lambda_0$ (and the fixed variance normalization of $T$).

*Lipschitz bound for inverse square-roots.* On $\mathcal{E}$, both $\Sigma_{XX}$ and $\widehat{\Sigma}_{XX}$ have eigenvalues bounded below by $\lambda_0/2$. The matrix function $A \mapsto A^{-1/2}$ is operator-Lipschitz on $[\lambda_0/2, \infty)$, implying

$$\|\widehat{S} - S\|_{\mathrm{op}} = \|\widehat{\Sigma}_{XX}^{-1/2} - \Sigma_{XX}^{-1/2}\|_{\mathrm{op}} \ \leq \ C_S \, \|\widehat{\Sigma}_{XX} - \Sigma_{XX}\|_{\mathrm{op}} \ \leq \ C_S \, \|\widehat{\Sigma} - \Sigma\|_{\mathrm{op}}, \tag{11}$$

where $C_S > 0$ depends only on $\lambda_0$ (e.g., one may take $C_S \asymp \lambda_0^{-3/2}$).

*Bound* $\|\widehat{\Omega}_M - \Omega_M\|_{\mathrm{op}}$. Apply equation 5 and submultiplicativity, using equation 7, equation 8, equation 9, equation 10, and equation 11. On $\mathcal{E}$,

$$\|\widehat{\Omega}_M - \Omega_M\|_{\mathrm{op}} \ \leq \ \|\widehat{S} - S\|_{\mathrm{op}} \|\widehat{M}\|_{\mathrm{op}} \|\widehat{S}\|_{\mathrm{op}} + \|S\|_{\mathrm{op}} \|\widehat{M} - M\|_{\mathrm{op}} \|\widehat{S}\|_{\mathrm{op}} + \|S\|_{\mathrm{op}} \|M\|_{\mathrm{op}} \|\widehat{S} - S\|_{\mathrm{op}} \ \leq \ C' \, \|\widehat{\Sigma} - \Sigma\|_{\mathrm{op}},$$

for a constant $C' > 0$ depending only on $\lambda_0, \Lambda_0$. This proves the second inequality in the theorem statement.

*Conclude and plug concentration.* Combining equation 4 with the previous display gives, on $\mathcal{E}$,

$$\|\sin\Theta(\widehat{U}, U_\star)\|_{\mathrm{op}} \ \leq \ \frac{C}{\gamma} \|\widehat{\Omega}_M - \Omega_M\|_{\mathrm{op}} \ \leq \ \frac{CC'}{\gamma} \|\widehat{\Sigma} - \Sigma\|_{\mathrm{op}}.$$

Finally, Theorem 2 yields $\|\widehat{\Sigma} - \Sigma\|_{\mathrm{op}} \lesssim K^2\big(\sqrt{(p + \log(1/\delta))/n} + (p + \log(1/\delta))/n\big)$, and for the usual regime $n \gtrsim p + \log(1/\delta)$ this implies the stated $\lesssim \frac{K^2}{\gamma}\sqrt{\frac{p+\log(1/\delta)}{n}}$ rate (absorbing constants into $\lesssim$). The projector inequality $\|\widehat{\Pi} - \Pi_\star\|_{\mathrm{op}} \leq 2\|\sin\Theta(\widehat{U}, U_\star)\|_{\mathrm{op}}$ follows from standard principal-angle identities. $\qquad\square$

*Proof of Theorem 4.* Write $X := X_M \in \mathbb{R}^p$. Under the Gaussian IB setting of Sec. 4.1 (jointly Gaussian $(X, T)$), the population conditional mean is linear:

$$g^\star(x) = \mathbb{E}[T \mid X = x] = \beta^\top x, \qquad \beta := \Sigma_{XX}^{-1}\Sigma_{XT},$$

(where we use $\mathbb{E}(Z) = 0$). Since $T$ is one-dimensional, the CCA/GIB operator $\Omega_M$ has rank at most 1, and the MB–GIB encoder/decoder produces the same predictor as ordinary least squares on $(X, T)$ (up to an irrelevant scaling of the latent). In particular, the empirical MB–GIB predictor $\widehat{g}$ is of the form

$$\widehat{g}(x) = \widehat{\beta}^\top x, \qquad \widehat{\beta} := \widehat{\Sigma}_{XX}^{-1}\widehat{\Sigma}_{XT},$$

where $\widehat{\Sigma}$ is the empirical covariance of $Z = (X^\top, T)^\top$ and $\widehat{\Sigma}_{XX}, \widehat{\Sigma}_{XT}$ are its blocks. (Equivalently: the top eigenvector of $\widehat{\Omega}_M$ is proportional to $\widehat{\Sigma}_{XX}^{-1/2}\widehat{\Sigma}_{XT}$; unwhitening yields $\widehat{W} \propto \widehat{\Sigma}_{XX}^{-1}\widehat{\Sigma}_{XT}$; the optimal linear decoder cancels the scaling.)

*Excess risk equals $L_2$ prediction error.* For squared loss, the Bayes rule $g^\star$ is the conditional mean, and the excess risk satisfies the identity

$$\mathcal{R}_{\mathrm{s}}(\widehat{g}) - \mathcal{R}_{\mathrm{s}}(g^\star) = \mathbb{E}\big[(\widehat{g}(X) - g^\star(X))^2\big] = \mathbb{E}\big[((\widehat{\beta} - \beta)^\top X)^2\big] = (\widehat{\beta} - \beta)^\top \Sigma_{XX}(\widehat{\beta} - \beta).$$

Using Assumption 2, $\|\Sigma_{XX}\|_{\mathrm{op}} \leq \Lambda_0$, hence

$$\mathcal{R}_{\mathrm{s}}(\widehat{g}) - \mathcal{R}_{\mathrm{s}}(g^\star) \ \leq \ \Lambda_0 \, \|\widehat{\beta} - \beta\|_2^2. \tag{12}$$

*Lipschitz control of $\widehat{\beta}$ in terms of covariance errors.* Let $\Delta_{XX} := \widehat{\Sigma}_{XX} - \Sigma_{XX}$ and $\Delta_{XT} := \widehat{\Sigma}_{XT} - \Sigma_{XT}$. Then

$$\widehat{\beta} - \beta = \widehat{\Sigma}_{XX}^{-1}\widehat{\Sigma}_{XT} - \Sigma_{XX}^{-1}\Sigma_{XT} = \widehat{\Sigma}_{XX}^{-1}\Delta_{XT} + \big(\widehat{\Sigma}_{XX}^{-1} - \Sigma_{XX}^{-1}\big)\Sigma_{XT}.$$

Work on the event

$$\mathcal{E} := \left\{\|\widehat{\Sigma} - \Sigma\|_{\mathrm{op}} \leq \frac{\lambda_0}{2}\right\}.$$

On $\mathcal{E}$, $\lambda_{\min}(\widehat{\Sigma}_{XX}) \geq \lambda_0/2$, hence

$$\|\widehat{\Sigma}_{XX}^{-1}\|_{\mathrm{op}} \leq \frac{2}{\lambda_0}, \qquad \|\Sigma_{XX}^{-1}\|_{\mathrm{op}} \leq \frac{1}{\lambda_0}, \qquad \|\widehat{\Sigma}_{XX}^{-1} - \Sigma_{XX}^{-1}\|_{\mathrm{op}} \leq \|\widehat{\Sigma}_{XX}^{-1}\|_{\mathrm{op}} \|\Delta_{XX}\|_{\mathrm{op}} \|\Sigma_{XX}^{-1}\|_{\mathrm{op}} \leq \frac{2}{\lambda_0^2} \|\Delta_{XX}\|_{\mathrm{op}}.$$

Moreover, $\|\Delta_{XX}\|_{\mathrm{op}} \leq \|\widehat{\Sigma} - \Sigma\|_{\mathrm{op}}$ and $\|\Delta_{XT}\|_{\mathrm{op}} \leq \|\widehat{\Sigma} - \Sigma\|_{\mathrm{op}}$ by the block-operator bound used in Theorem 2. Finally, since the block covariance $\left( \begin{smallmatrix} \Sigma_{XX} & \Sigma_{XT} \\ \Sigma_{TX} & \Sigma_{TT} \end{smallmatrix} \right) \succeq 0$, its Schur complement gives $\Sigma_{XT} \Sigma_{TT}^{-1} \Sigma_{TX} \preceq \Sigma_{XX}$. Rescaling $T$ so that $\Sigma_{TT} = 1$ (w.l.o.g. for a scalar $T$, absorbing this into $K$), we obtain $\|\Sigma_{XT}\|_{\mathrm{op}}^2 = \|\Sigma_{XT} \Sigma_{TX}\|_{\mathrm{op}} \leq \|\Sigma_{XX}\|_{\mathrm{op}} \leq \Lambda_0$, hence $\|\Sigma_{XT}\|_{\mathrm{op}} \leq \sqrt{\Lambda_0}$. Combining the above, on $\mathcal{E}$,

$$\|\widehat{\beta} - \beta\|_2 \ \leq \ \|\widehat{\Sigma}_{XX}^{-1}\|_{\mathrm{op}} \|\Delta_{XT}\|_{\mathrm{op}} + \|\widehat{\Sigma}_{XX}^{-1} - \Sigma_{XX}^{-1}\|_{\mathrm{op}} \|\Sigma_{XT}\|_{\mathrm{op}} \ \leq \ \left( \frac{2}{\lambda_0} + \frac{2\sqrt{\Lambda_0}}{\lambda_0^2} \right) \|\widehat{\Sigma} - \Sigma\|_{\mathrm{op}}.$$

Therefore,

$$\|\widehat{\beta} - \beta\|_2^2 \ \leq \ C_\beta \|\widehat{\Sigma} - \Sigma\|_{\mathrm{op}}^2, \qquad C_\beta := \left( \frac{2}{\lambda_0} + \frac{2\sqrt{\Lambda_0}}{\lambda_0^2} \right)^2, \tag{13}$$

where $C_\beta$ depends only on $\lambda_0, \Lambda_0$.

*Conclude the operator-norm-squared bound.* Plugging equation 13 into equation 12 yields, on $\mathcal{E}$,

$$\mathcal{R}_{\mathrm{s}}(\widehat{g}) - \mathcal{R}_{\mathrm{s}}(g^\star) \ \leq \ \Lambda_0 \, C_\beta \|\widehat{\Sigma} - \Sigma\|_{\mathrm{op}}^2 \ = \ C'' \|\widehat{\Sigma} - \Sigma\|_{\mathrm{op}}^2,$$

with $C'' := \Lambda_0 C_\beta$, depending only on $\lambda_0, \Lambda_0$, as claimed.

*Convert to an explicit rate via covariance concentration.* By Theorem 2, with probability at least $1 - \delta$,

$$\|\widehat{\Sigma} - \Sigma\|_{\mathrm{op}} \ \lesssim \ K^2 \left( \sqrt{\frac{p + 1 + \log(1/\delta)}{n}} + \frac{p + 1 + \log(1/\delta)}{n} \right).$$

Squaring and using the usual regime $n \gtrsim p + \log(1/\delta)$ gives

$$\mathcal{R}_{\mathrm{s}}(\widehat{g}) - \mathcal{R}_{\mathrm{s}}(g^\star) \ \lesssim \ K^4 \frac{p + \log(1/\delta)}{n},$$

after absorbing constants into $\lesssim$. This completes the proof. $\qquad\square$

*Proof of Corollary 1.* For squared loss $\ell(t, \hat{t}) = (t - \hat{t})^2$, the Bayes rule is the conditional mean. Under MB invariance $p_{\mathrm{s}}(T \mid X_M) = p_{\mathrm{t}}(T \mid X_M)$, the conditional mean is shared: $g^\star(x_M) = \mathbb{E}_{p_{\mathrm{s}}}[T \mid X_M = x_M] = \mathbb{E}_{p_{\mathrm{t}}}[T \mid X_M = x_M]$, and is target-optimal among predictors measurable w.r.t. $X_M$.

For any domain $\nu \in \{\mathrm{s}, \mathrm{t}\}$, the regression decomposition yields

$$\mathcal{R}_\nu(\widehat{g}) - \mathcal{R}_\nu(g^\star) = \mathbb{E}_{p_\nu(X_M)}\big[(\widehat{g}(X_M) - g^\star(X_M))^2\big],$$

which proves the stated conditional excess-risk identity.

If $p_{\mathrm{t}} \ll p_{\mathrm{s}}$ on $X_M$, then by change of measure,

$$\mathcal{R}_{\mathrm{t}}(\widehat{g}) - \mathcal{R}_{\mathrm{t}}(g^\star) = \mathbb{E}_{p_{\mathrm{s}}(X_M)}\left[(\widehat{g}(X_M) - g^\star(X_M))^2 \frac{p_{\mathrm{t}}(X_M)}{p_{\mathrm{s}}(X_M)}\right] \leq \left( \operatorname*{ess\,sup}_{x_M} \frac{p_{\mathrm{t}}(x_M)}{p_{\mathrm{s}}(x_M)} \right) \mathbb{E}_{p_{\mathrm{s}}(X_M)}\big[(\widehat{g}(X_M) - g^\star(X_M))^2\big].$$

The last expectation equals $\mathcal{R}_{\mathrm{s}}(\widehat{g}) - \mathcal{R}_{\mathrm{s}}(g^\star)$ by the same decomposition, yielding

$$\mathcal{R}_{\mathrm{t}}(\widehat{g}) - \mathcal{R}_{\mathrm{t}}(g^\star) \leq \rho_M\big(\mathcal{R}_{\mathrm{s}}(\widehat{g}) - \mathcal{R}_{\mathrm{s}}(g^\star)\big), \qquad \rho_M := \operatorname*{ess\,sup}_{x_M} \frac{p_{\mathrm{t}}(x_M)}{p_{\mathrm{s}}(x_M)}.$$

Finally, applying Theorem 4 gives, with probability at least $1 - \delta$,

$$\mathcal{R}_{\mathrm{t}}(\widehat{g}) - \mathcal{R}_{\mathrm{t}}(g^\star) \ \lesssim \ \rho_M K^4 \frac{p + \log(1/\delta)}{n}.$$

If $p_{\mathrm{s}}(X_M) = p_{\mathrm{t}}(X_M)$, then $\rho_M = 1$ and the source and target excess risks coincide. $\qquad\square$

*Proof of Corollary 2.* For squared loss, the regression decomposition implies that for any domain $\nu \in \{s, t\}$ and any predictor $g$,

$$\mathcal{R}_\nu(g) - \mathcal{R}_\nu(g_\nu^\star) = \mathbb{E}_{p_\nu(X_M)}\big[(g(X_M) - g_\nu^\star(X_M))^2\big], \tag{14}$$

where $g_\nu^\star(x_M) = \mathbb{E}_{p_\nu}[T \mid X_M = x_M]$. Applying equation 14 with $\nu = t$ and $g = \widehat{g}$ yields

$$\mathcal{R}_t(\widehat{g}) - \mathcal{R}_t(g_t^\star) = \mathbb{E}_{p_t(X_M)}\big[(\widehat{g}(X_M) - g_t^\star(X_M))^2\big].$$

Now write $g_t^\star = g_s^\star + \Delta$, so that pointwise in $x_M$,

$$\widehat{g} - g_t^\star = (\widehat{g} - g_s^\star) - \Delta.$$

Expanding the square gives

$$(\widehat{g} - g_t^\star)^2 = (\widehat{g} - g_s^\star)^2 - 2(\widehat{g} - g_s^\star)\Delta + \Delta^2,$$

hence

$$(\widehat{g} - g_t^\star)^2 - (\widehat{g} - g_s^\star)^2 = -2(\widehat{g} - g_s^\star)\Delta + \Delta^2.$$

Taking absolute values and applying $|ab| \leq |a||b|$ yields the pointwise bound

$$\big|(\widehat{g} - g_t^\star)^2 - (\widehat{g} - g_s^\star)^2\big| \leq 2\,|\widehat{g} - g_s^\star|\,|\Delta| + \Delta^2.$$

Finally, taking expectation under the target marginal $p_t(X_M)$ gives

$$\left|\mathbb{E}_{p_t(X_M)}\big[(\widehat{g} - g_t^\star)^2\big] - \mathbb{E}_{p_t(X_M)}\big[(\widehat{g} - g_s^\star)^2\big]\right| \leq 2\,\mathbb{E}_{p_t}\big[|\widehat{g}(X_M) - g_s^\star(X_M)|\,|\Delta(X_M)|\big] + \mathbb{E}_{p_t}\big[\Delta(X_M)^2\big].$$

Substituting $\mathbb{E}_{p_t}[(\widehat{g} - g_t^\star)^2] = \mathcal{R}_t(\widehat{g}) - \mathcal{R}_t(g_t^\star)$ from equation 14 proves the claimed inequality.

If shifts occur only outside $M$, then $p_s(T \mid X_M) = p_t(T \mid X_M)$ and thus $\Delta \equiv 0$, which yields the stated exact identity. $\qquad\square$

