# OpenReview forum: "Causally-Aware Information Bottleneck for Domain Adaptation"
_TMLR — Accepted by TMLR_

### Review · Reviewer_nkM8 · 2026-03-05

**Summary Of Contributions:**

This paper studies domain adaptation when the target variable $T$ is observed in a labeled source domain but entirely missing in an unlabeled target domain. The proposed approach uses local causal structure around $T$, the Markov blanket $MB(T)$ (or the causal parents), to restrict the inputs to an Information Bottleneck (IB) model. In the linear Gaussian case, the method yields a closed-form Gaussian IB solution closely related to CCA (MB--GIB). For nonlinear or non-Gaussian settings, the authors propose a variational IB encoder-decoder (MB-VIB) trained on source and deployed zero-shot to target. The key stability assumption is $p_s(T\mid MB(T))=p_t(T\mid MB(T))$ ("MB-invariance"), under which the Bayes predictor based on $MB(T)$ transfers.

**Audience:**

No

**Audience Explanation:**

However, I think my major concerns is that the methods proposed might lack novelty or broad applicability. In particular:

---
### 1) Significance and novelty relative to existing invariance ideas

At a high level, the key message seems to be: if $T \perp X\setminus(MB(T)\cup\{T\}) \mid MB(T)$ and $p_s(T\mid MB(T))=p_t(T\mid MB(T))$, then predictions based on $MB(T)$ transfer. This is conceptually close to standard Markov blanket sufficiency and invariance arguments: once $MB(T)$ is known and stable, using it for robust prediction is nearly the canonical solution.

I therefore struggled to assess whether the main contribution is algorithmic novelty, a new guarantee, or mainly a packaging of well-known causal ideas with IB. In particular:
- For the IB case: Under the stated conditional independence, the population predictor is simply $g^\star(x_M)=\mathbb{E}[T\mid X_M=x_M]$. The benefit of the IB objective over directly estimating $g^\star$ is not fully convincing as a core contribution unless the paper clarifies what additional robustness/regularization IB provides beyond standard supervised learning on $MB(T)$.
- The Gaussian theory (CCA equivalence, lossless restriction) is elegant, but it also appears to be a fairly direct consequence of Gaussian conditional independence and standard CCA/IB connections.

---
### 2) Main technical contributions

Theorems 1 to 3 appear to follow standard patterns. If these are intended to be main contributions, the paper should clarify what is nontrivial or new in these results relative to existing literature on CCA/IB finite-sample analysis.

Moreover, MB-VIB is described as a practical extension, but the theoretical discussion for nonlinear/non-Gaussian settings is mainly qualitative. If the paper's main value is theory, the nonlinear case feels underdeveloped; if the main value is practical transfer, then the theory might be better framed as supportive rather than central.

---
### 3) Practicality of the Markov blanket assumption

The method requires access to $MB(T)$ (or an estimate thereof). In many realistic applications, reliably estimating $MB(T)$ from observational data is difficult, especially with finite samples, hidden confounding, measurement error, and model misspecification.

The current text briefly mentions that one can "reliably estimate" the blanket, but it does not provide sufficient guidance, failure analysis, or empirical evidence that the overall method remains effective when $MB(T)$ is estimated with error. A more encompassing discussion on the estimation/learning problem of $MB(T)$ would make the paper more meaty.

---
### 4) Clarity and notation

Several notational choices were confusing:
- Using $T$ as the target variable while also using $t$ as an index for the target domain and writing $T=X_t$ is easy to misread. Additionally, notations like $X^{(j)}_{t-1,t}$ are hard to understand without a very explicit explanation of the notation. It is not always clear whether $t$ is fixed (domain label) or varies (node index).
- The paper uses $S$ in multiple roles (spurious block in the motivating example; also $X_S$ as an input scope), and it uses both $X_M$ and $X_S$ to denote the restricted inputs. This makes it unclear whether $X_S$ equals the full $MB(T)$, parents only, or a chosen subset.

---

**Broader Impact Concerns:**

None.

**Claims And Evidence:**

Yes

**Claims Explanation:**

The paper has some clear strengths. For instance:

- It presents a clean, easy, and lightweight pipeline for zero-shot imputation.
- The Markov blanket restriction is intuitive and natural under MB-invariance.
- The Gaussian case yields an explicit and computationally cheap procedure (CCA-style), which can be attractive in high-dimensional graphs when $|MB(T)|$ is small.
- The empirical section covers both synthetic SEMs and real data, which I appreciate.

**Requested Changes:**

I suggest the following changes to strengthen the paper from its current form.

The authors should:
- Sharpen the claim of novelty, explicitly position what is new beyond "restrict to $MB(T)$", and include a baseline that directly predicts $T$ from $MB(T)$ (e.g., linear regression / kernel ridge / MLP restricted to $MB(T)$) to isolate what IB adds.
- Clarify which precise claims are the core contribution, and calibrate the presentation accordingly (either strengthen novelty of the theory, or reframe the theory as standard but useful justification for a practical method).
- Add a clearer algorithmic discussion: how exactly do you propose to estimate $MB(T)$ in practice, and under what conditions do you expect it to be reliable?
- Move at least one sensitivity experiment on blanket misspecification into the main paper (not only supplement), since blanket availability is central to the method.
- The authors should adopt a consistent convention such as  $e\in\\{s,t\\}$ for environment/domain, $M:=MB(T)$ and always use $X_M$ (avoid $X_S$ unless $S$ is a well-defined set distinct from $M$). If you allow "parents-only" scope, denote it by $P:=Pa(T)$ and use $X_P$.
- Table 1 and Fig. 2 are meant to motivate the difference between global GIB and MB-GIB, but the formal definition of these approaches appears later. I suggest either:
  (i) moving the formal introduction of MB-GIB and global GIB earlier (a short preliminaries section), or
  (ii) keeping the early motivation more high-level and deferring detailed visualizations until after the method is defined.
- When claiming robustness, it would help to explicitly distinguish robustness to shifts outside $MB(T)$ (covered by MB-invariance) from cases where the mechanism $p(T\mid MB(T))$ itself drifts (which breaks the guarantees).

---

> ### Author Response · Authors · 2026-04-19
>
> We thank the reviewer for the careful, balanced, and detailed feedback. We agree that notation and presentation need cleanup and will treat this as a revision priority. In particular, we will adopt a consistent convention for the target variable versus the domain or environment index, and for blanket-restricted inputs, and we will clarify throughout whether the scope is parents-only or the full Markov blanket MB(T).
>
> We also agree with the core concern that the current draft does not yet separate clearly enough what is new in the paper from what follows directly from standard Markov blanket sufficiency and invariance reasoning. In the revision, we will sharpen the novelty claim substantially. Specifically, we will no longer present "predict from MB(T) under invariance" itself as the novelty; instead, we will emphasize the narrower contributions: a closed-form MB-restricted Gaussian IB solution with a lossless restriction result in the linear-Gaussian case, and a practical MB-VIB formulation for nonlinear and non-Gaussian settings. We will also explicitly distinguish robustness to shifts outside MB(T), which is covered by the MB-invariance assumption, from mechanism drift in p(T | MB(T)), which breaks the guarantee — a distinction that is conceptually important and currently underemphasized. We will also calibrate the theory claims more carefully, presenting part of the finite-sample analysis as supportive theory for the proposed estimator rather than as the primary source of novelty.
>
> We have now added the requested direct baselines that predict T from MB(T) without an IB objective, using source-side validation tuning. Results on the MAGIC-IRRI benchmark are as follows:
>
> | Method | MAE | RMSE | R² |
> |---|---|---|---|
> | LinearReg (MB only) | 5.580 | 7.019 | 0.569 |
> | Ridge (MB only, tuned) | 5.580 | 7.019 | 0.569 |
> | KernelRidge (MB only, tuned) | 5.659 | 7.112 | 0.558 |
> | MLP (MB only, tuned) | 6.795 | 8.825 | 0.318 |
> | MB-GIB | 5.571 | 7.008 | 0.567 |
> | MB-VIB | 7.084 | 10.019 | 0.121 |
> | Bayesian Network | 9.383 | 11.187 | −0.096 |
> | Pure DNN (unrestricted) | 14.452 | 17.791 | −1.771 |
>
> These results confirm that in the linear-Gaussian MAGIC-IRRI setting, OLS on MB(T) is already near-optimal and MB-GIB provides no additional benefit over a plain linear baseline, which is expected from theory since MB-GIB reduces to CCA in this regime. We acknowledge this directly. We also note that the tuned MLP on MB(T) outperforms MB-VIB on this benchmark, which further supports our revised framing: the variational bottleneck can over-compress in well-specified linear-Gaussian settings, and IB's primary benefit lies in nonlinear and non-Gaussian regimes where closed-form estimation is unavailable. Across all MB-restricted methods, the blanket restriction itself is the primary driver of robustness, with all of them substantially outperforming the unrestricted Bayesian Network and Pure DNN. We will revise the paper to frame the contributions accordingly: blanket restriction as the core transfer mechanism, and IB as a principled estimator that provides regularization and extends to nonlinear and non-Gaussian settings where closed-form estimation is unavailable.
>
> In parallel, we will expand the practical discussion of how MB(T) is obtained in practice by adding a concrete paragraph on local blanket discovery and conditional-independence-test-based pruning, together with expected failure modes such as hidden confounding, finite-sample instability, measurement error, and model misspecification. We will also move a blanket-misspecification sensitivity experiment into the main paper. All of these revisions are already underway.
>
> Finally, we will revise the early presentation so that the formal definitions of global GIB and MB-GIB appear before the more detailed motivating comparisons. Thank you again for these concrete and very helpful suggestions.

---

### Review · Reviewer_MRWV · 2026-03-11

**Summary Of Contributions:**

Sure — here is a review-style draft you can paste:

The paper studies a domain-adaptation setting in which the target variable is observed in the source domain but completely missing at deployment in the target domain, and proposes to impute that target using a causally restricted information bottleneck. The main idea is to build the representation only from the parents or Markov blanket of the target, under the assumption that the conditional mechanism $p_s(T\mid MB(T)) = p_t(T\mid MB(T))$ remains invariant across domains. On the methodological side, the paper introduces two variants: MB-GIB, a closed-form Gaussian Information Bottleneck for the linear-Gaussian case that reduces to a CCA-style projection, and MB-VIB, a variational nonlinear/non-Gaussian extension with flexible decoders. In the Gaussian MB–GIB regime, the paper proves that restricting the bottleneck to the Markov blanket is lossless relative to using all non-target variables, that the population conditional predictor transfers zero-shot under MB-invariance, and that the empirical MB–GIB estimator satisfies finite-sample concentration and excess-risk guarantees; for MB–VIB, the paper gives a population-level justification but not equally strong non-asymptotic theory.

Overall, the paper’s contribution is a simple and coherent bridge between causal structure and bottleneck learning for target imputation under shift.

Key strengths: the paper addresses a clear and practically relevant problem; the core idea of restricting the bottleneck to the target’s Markov blanket is intuitive and well aligned with causal transfer; the Gaussian theory is the strongest part of the submission, especially the lossless blanket-restriction result and the zero-shot transfer argument.

Key weaknesses: the guarantees rely heavily on the correctness of the Markov blanket and on the MB-invariance assumption, so the method may be fragile when the blanket is misspecified or when the target mechanism itself changes; the nonlinear MB-VIB side is less theoretically developed than the Gaussian MB-GIB case; and some supporting ingredients, such as the concentration analysis, appear more standard than novel.

**Audience:**

Yes

**Audience Explanation:**

Yes, I believe at least part of the TMLR audience would be interested in these findings. The paper sits at the intersection of domain adaptation, causality, and representation learning, and proposes a fairly clear idea: combining Markov blanket–based variable selection with information bottleneck methods for target imputation under shift. That is a topic that should be relevant to readers working on robust prediction, transfer under distribution shift, causally informed machine learning, and missing-target or label-scarce settings.

**Broader Impact Concerns:**

No major broader-impact concerns stood out to me that would by themselves require an extensive dedicated Broader Impact Statement. The paper is primarily a methodological contribution on domain adaptation and target imputation under distribution shift, and its most immediate impact appears to be technical rather than socially sensitive

**Claims And Evidence:**

Yes

**Claims Explanation:**

While I am not fully expert in this specific sub-area, and it is possible that I may have missed or misunderstood some technical details, my overall impression is that the main claims are supported reasonably well by the combination of theory and experiments. In particular, the paper gives a clear formal treatment of the linear-Gaussian MB-GIB setting, including the claim that restricting the encoder to the Markov blanket is lossless in that regime, as well as finite-sample and transfer guarantees under the MB-invariance assumption.

That said, I think the evidence is strongest for the Gaussian/linear MB-GIB part of the paper. The nonlinear MB-VIB variant appears more practically motivated and empirically validated than fully established theoretically, and some of the broader transfer claims depend critically on the MB-invariance assumption and on having the correct Markov blanket. The paper does acknowledge these limitations explicitly.

**Requested Changes:**

1. Please point more clearly in the main text to the precise theorem statements that appear in the appendix. I found it difficult at times to identify exactly where the main formal results corresponding to Sections 4.1 and 4.2 are stated and proved. In particular, these results are not easy to locate in Appendix A in their current form. The appendix feels somewhat diffuse, in the sense that the relevant proofs are present, but they are embedded among many additional intermediate arguments and technical details, which makes it hard for the reader to quickly match the claims in the main text to their formal justification. I believe the paper would benefit from more explicit signposting in the main text, for example by directly referring to the corresponding theorem/proposition numbers whenever a theoretical claim is introduced.

2. Please state the appendix theorems explicitly before their proofs. In several places, the appendix moves too quickly into proof material without first presenting a clean standalone statement of the corresponding theorem, proposition, or lemma. I think readability would improve substantially if each proof were preceded by the precise result being established. This would make the logical structure of the appendix much easier to follow and would also help connect the theory back to the claims made in Sections 4.1 and 4.2.

3. I would encourage the authors to revise the proof presentation style, in particular the use of “Step 1”, “Step 2”, etc., which currently feels somewhat informal and stylistically unusual for this type of presentation. In its current form, this proof style reads a bit mechanically and gives the appendix a somewhat less polished feel than the rest of the paper.

---

> ### Author Response · Authors · 2026-04-19
>
> We sincerely thank the reviewer for the thoughtful and positive assessment. We especially appreciate the recognition that the strongest part of the paper is the Gaussian MB-GIB analysis and the clear identification of the paper's main limitations, particularly the reliance on MB correctness and MB-invariance. We agree with this characterization and will make the scope and assumptions more explicit in the revision.
>
> We also appreciate the concrete feedback on presentation. We agree that the appendix currently makes it harder than necessary to map the claims in Sections 4.1–4.4 to their exact formal statements. In the revision, we will add direct theorem and proposition references in the main text whenever a theoretical claim is made, and we will reorganize the appendix so that each theorem, proposition, or lemma is stated cleanly before its proof. We will also revise the proof style to remove the current "Step 1 / Step 2" presentation in favor of a more standard and polished format. These changes are already underway and will be reflected in the revision.
>
> More broadly, your comments helped us see that even when the technical content is present, signposting matters for credibility and readability. We will treat this as a substantive revision priority. Thank you again for the careful reading and constructive suggestions.

---

### Review · Reviewer_AWeU · 2026-04-09

**Summary Of Contributions:**

**Summary**

This paper addresses unsupervised domain adaptation under the assumption that the causal graph (specifically, the Markov Blanket of the target) is known.
Based on the MB invariance assumption, which assumes that the conditional distribution of the target given the MB remains fixed even if the marginal distributions change, the authors propose MB-GIB (using CCA) for linear cases and MB-VIB (using info bottleneck) for non-linear cases, along with theoretical error bounds.

**Strengths**

(1) The paper is well-organized and the overall logic is easy to follow.
(2) The experiments cover a good mix of scenarios, from synthetic structural equation models to real-world benchmarks.

**Weaknesses**

(1) From my perspective, the strong assumptions somewhat trivialize the UDA problem. The core challenge of UDA is figuring out how to separate domain-invariant features from domain-specific noise without target labels. By assuming we already know the exact causal graph and that MB-invariance strictly holds, this main challenge is completely bypassed. The problem essentially degrades into a standard "prevent overfitting to the source domain" task.

(2) The theoretical analysis lacks DA-specific insights. Following up on W1, since the task reduces to regularizing the source model, the provided theoretical guarantees (like the finite-sample error bounds) just look like standard concentration bounds for regularized regression. They offer limited new insights unique to domain adaptation (e.g., bounding target risk using source risk and domain divergence).

(3) Limited baseline comparisons. The experimental comparison is mostly limited to basic models (ERM, pure DNN) and just one invariant learning baseline (IIB-style). To make the results convincing, the authors need to compare against a broader set of mainstream OOD or invariant learning methods.

(4) Sensitivity to incorrect causal graphs. Causal discovery is a notoriously difficult task, especially when dealing with high-dimensional data. Yet, the whole framework heavily relies on having the exact correct Markov Blanket. In practice, causal discovery algorithms are noisy, and obtaining a perfect DAG is highly unrealistic. The paper is missing a critical ablation study showing how the model performs if the assumed MB is slightly misspecified (e.g., missing a true parent node or accidentally including a spurious variable).

(5) The actual contribution of the Information Bottleneck is not very clear and specific. The authors argue that VIB is needed to stop the network from overfitting to the source marginal distributions. However, since the input is already restricted to just the MB variables, it is unclear if the complex VIB formulation is actually better than simply applying standard neural network regularizations (like strong dropout or weight decay).

**Audience:**

Yes

**Audience Explanation:**

DA is a classic and relevant problem in machine learning, and it falls squarely within the scope of TMLR

**Claims And Evidence:**

No

**Claims Explanation:**

The submission lacks a sufficient, targeted analysis to justify the irreplaceability of the proposed Information Bottleneck module.

**Requested Changes:**

I strongly encourage the authors to reposition their core contribution. Instead of framing this as a solution to standard UDA, it would be much more compelling to explicitly introduce "Domain Adaptation under a Known Causal Graph" as a novel problem setting.
The authors should thoroughly discuss the practical significance and real-world applicability of this specific setting (e.g., when and why would a causal graph be perfectly known in practice?). Explicitly justifying this setting would significantly strengthen the fundamental rationale for the paper's existence. Furthermore, the authors must validate their claimed conclusions through highly targeted experiments (e.g., direct ablations isolating the IB module, and sensitivity tests on graph misspecification) rather than relying on generic UDA benchmarks.

---

> ### Author Response · Authors · 2026-04-19
>
> We thank the reviewer for the careful reading and constructive feedback. We agree with the central point that the current draft did not position the problem setting sharply enough. Our intent is not to claim a solution to generic unsupervised domain adaptation, but rather to study domain adaptation with local causal knowledge, where the target's parents or Markov blanket are known or can be estimated and the conditional mechanism p(T | MB(T)) is stable across domains. We will revise the framing accordingly and better motivate when this setting is practically meaningful.
>
> We also agree that the current draft does not yet isolate clearly enough what the Information Bottleneck contributes beyond simply predicting from MB(T). We have now added properly tuned direct baselines that predict T from MB(T) without any IB objective, using source-side validation for hyperparameter selection. Results on the MAGIC-IRRI benchmark are as follows:
>
> | Method | MAE | RMSE | R² |
> |---|---|---|---|
> | LinearReg (MB only) | 5.580 | 7.019 | 0.569 |
> | Ridge (MB only, tuned) | 5.580 | 7.019 | 0.569 |
> | KernelRidge (MB only, tuned) | 5.659 | 7.112 | 0.558 |
> | MLP (MB only, tuned) | 6.795 | 8.825 | 0.318 |
> | MB-GIB | 5.571 | 7.008 | 0.567 |
> | MB-VIB | 7.084 | 10.019 | 0.121 |
> | Bayesian Network | 9.383 | 11.187 | −0.096 |
> | Pure DNN (unrestricted) | 14.452 | 17.791 | −1.771 |
>
> These results directly confirm the reviewer's intuition for the linear-Gaussian setting: when the conditional mean is linear, OLS restricted to MB(T) is already near-optimal, and MB-GIB, which reduces to CCA in this regime, provides no additional benefit over a plain linear baseline. We acknowledge this directly. We also note that the tuned MLP on MB(T) outperforms MB-VIB on this benchmark, which further supports our revised framing: the variational bottleneck can over-compress in well-specified linear-Gaussian settings, and IB's primary benefit lies in nonlinear and non-Gaussian regimes where closed-form estimation is unavailable.
>
> The results also sharpen the picture of where IB adds value and where it does not. The blanket restriction itself is the primary driver of transfer robustness, as evidenced by the large gap between all MB-restricted methods and the unrestricted Bayesian Network and Pure DNN. Within MB-restricted methods, the IB objective does not improve over linear regression in the linear-Gaussian regime, which is consistent with theory. Our intended claim is therefore not that IB is universally required once MB(T) is known, but rather that it provides a principled compression and regularization mechanism that extends naturally to nonlinear and non-Gaussian settings where a closed-form conditional estimator is unavailable. We will revise the paper to frame the contributions accordingly: blanket restriction as the core transfer mechanism, and IB as a principled estimator that extends this to flexible model classes. These revisions are already underway.
>
> Likewise, we agree that robustness to blanket misspecification is central. The paper already notes that performance can degrade under misspecified blankets and that misspecification ablations were conducted in the supplement, but we agree that at least one such sensitivity study should appear in the main paper.
>
> Finally, we accept the reviewer's point that some of the current theory reads more like regularized source-learning analysis than DA-specific theory. In the revision, we will reframe the Gaussian finite-sample results more clearly as estimator-specific guarantees for MB-GIB, while foregrounding the DA-specific message: the transfer guarantee comes from the combination of blanket restriction and MB-invariance, and the finite-sample theory quantifies how well the source estimator recovers that transferable predictor. We appreciate this suggestion and believe it will substantially improve the paper.

---

### Decision · Action_Editor_AJYs · 2026-05-20

**Recommendation:** Accept as is

**Additional Comments:**

This paper studies domain adaptation under the assumption that the target variable’s Markov blanket is known and invariant across domains, and proposes information bottleneck–based methods for imputing missing target variables under distribution shift. Reviewers agree that the paper is clearly written, technically coherent, and addresses a relevant problem at the intersection of causality and transferability. Multiple reviewers raise concerns that the strong assumptions simplify the domain adaptation problem and limit practical applicability, making the overall novelty appear incremental relative to existing invariance-based approaches.
Overall, the paper is viewed as a solid and well-executed contribution with useful insights for causally informed domain adaptation, though its impact is somewhat constrained by its reliance on strong structural assumptions. I recommend acceptance of the paper to TMLR but don't recommend it for Journal-to-conference track.

**Audience:**

Yes

**Audience Explanation:**

The results are interesting to researchers in domain adaptation and causal machine learning.

**Claims And Evidence:**

Yes

**Claims Explanation:**

Yes, the claims are supported by theoretical and empirical evidence.